# Machine learning-based health environmental-clinical risk scores in European children

## Abstract

**Background** Early life environmental stressors play an important role in the development of multiple chronic disorders. Previous studies that used environmental risk scores (ERS) to assess the cumulative impact of environmental exposures on health are limited by the diversity of exposures included, especially for early life determinants. We used machine learning methods to build early life exposome risk scores for three health outcomes using environmental, molecular, and clinical data.

**Methods** In this study, we analyzed data from 1622 mother-child pairs from the HELIX European birth cohorts, using over 300 environmental, 100 child peripheral, and 18 mother-child clinical markers to compute environmental-clinical risk scores (ECRS) for child behavioral difficulties, metabolic syndrome, and lung function. ECRS were computed using LASSO, Random Forest and XGBoost. XGBoost ECRS were selected to extract local feature contributions using Shapley values and derive feature importance and interactions.

**Results** ECRS captured 13%, 50% and 4% of the variance in mental, cardiometabolic, and respiratory health, respectively. We observed no significant differences in predictive performances between the above-mentioned methods. The most important predictive features were maternal stress, noise, and lifestyle exposures for mental health; proteome (mainly IL1B) and metabolome features for cardiometabolic health; child BMI and urine metabolites for respiratory health.

**Conclusions** Besides their usefulness for epidemiological research, our risk scores show great potential to capture holistic individual level non-hereditary risk associations that can inform practitioners about actionable factors of high-risk children. As in the post-genetic era personalized prevention medicine will focus more and more on modifiable factors, we believe that such integrative approaches will be instrumental in shaping future healthcare paradigms.

## Plain language summary

Growing up in different environments can greatly affect children's health later in life. This research looked at how living in cities, being exposed to chemicals, and other experiences before birth and during childhood, work together to influence children's mental, cardiovascular and respiratory health. We used advanced computer programs to help us understand these effects and estimate health risk scores. These scores are simple numerical measures that help us quantify the likelihood of children developing health issues based on their environmental exposures. Using those scores, the study identified key factors impacting children's health, in particular psycho-social, perceived environmental and prenatal pollutant exposures for mental health. It also revealed complex patterns and interactions between environmental factors. The results highlighted the potential of such risk scores to support the identification of actionable factors in high-risk children, informing tailored prevention measures in healthcare.

The environment (i.e., non-genetic factors), is a key driver of physical and mental health, studied for a long time in the literature in both adults and children[1,2]. It has been shown that early exposure to adverse environmental agents during sensitive periods such as pregnancy, birth, and childhood can have a long-term impact on both animal[3] and human health[4,5]. For instance, while most mental disorders begin during adolescence and early adulthood (e.g., schizophrenia, psychosis), their onsets are preceded by a prodromal phase with deviance from the typical neurodevelopmental trajectory taking place before clinical psychiatric symptoms. Not only important early life

events such as obstetric complications at birth but also the environment in which a child grows up, including factors such as lifestyle, family social capital and pollutant exposure, can influence their development as we showed in a recent exposome study of child behavioural problems[6].

In recent decades, the study of the impact of environmental exposure on health has evolved from analyzing individual factors to the broader analysis of the exposome. Described as "*the totality of human environmental exposures from conception onwards*"[7], the exposome recognizes that individuals are simultaneously exposed to a multitude of environmental factors

✉e-mail: lea.maitre@isglobal.org

and takes a holistic approach to the discovery of etiological factors for disease. Its main advantage over traditional "one-exposure-one-disease" approaches is that it provides a conceptual framework for investigating multiple environmental hazards (e.g., urban, chemical, lifestyle, social) and their combined effects. Classical single exposure analyses may be limited as the studied exposure association could arise from another correlated factor not taken into account and are, moreover, unable to capture interactions or cumulative effects from the exposure mixture. Exposures are not isolated, can be correlated with one another, and are likely to interact both among themselves (ExE) and with genetics (GxE) to drive health and non-communicable diseases[8,9]. Therefore, there is a need to collect a wide range of environmental data coupled with clinical biomarkers to comprehensively capture the main domains of the exposome before the apparition of clinical symptoms and apply more advanced modeling approaches suitable for complex mixtures of exposures that can leverage their interactions to facilitate its integration in observational and clinical studies.

Inspired by risk prediction models, such as the Framingham risk score for coronary heart disease[10] or polygenic risk scores (PRS)[11], environmental risk scores (ERS) are summary measures of the effects of multiple exposures used to estimate exposomic liability for a particular health outcome at an individual level[12]. Unlike genetics, some environmental factors are actionable, which gives ERS a broader potential for shaping public health policies by identifying actionable key factors and facilitating the implementation of preventive and personalized medicine measures. Combined with PRS, these scores could also serve as an initial step in identifying at-risk populations, who can then be directed to more specific clinical diagnostic tests, thereby serving as a complementary tool in the healthcare decision-making process[13,14]. ERS are usually built as a weighted sum of the individual exposure estimates, obtained either from meta-analysis or derived through linear regression models (from single multivariate models or several models, adjusted for multiple testing)[15]. This scheme, however, assumes that each environmental stressor individually acts in a linear dose-response relationship, while previous research has shown that their combined effect does not necessarily follow this rule[16,17].

The availability of rich data on multiple levels of environmental exposures in birth cohorts presents an opportunity to address the gap in large-scale studies on the association between the exposome and child and adolescent development. Previous ERS studies have been limited by the number of exposure variables or domains included[10,18,19]. In contrast, our study identify predictive environmental-clinical risk scores (ECRS) based on a wide array of pregnancy and childhood environmental exposures related to both external (e.g., air quality, lifestyle, psychosocial) and internal (e.g., blood metals, pesticides) factors and (pre)clinical factors (metabolites, proteins, co-morbidities), and link these to a range of physical and mental symptoms in the large European Human Early-Life Exposome (HELIX) cohort[20,21]. In the context of this study, we use the term prediction to refer to the inference of diagnostical risk scores that include pregnancy and childhood cross-sectional epidemiological factors to predict childhood liabilities at a single point in time.

In this work, we provide ECRS for three general health outcomes in children: a P-factor score derived from the Child Behavior CheckList (CBCL) for mental health, a metabolic syndrome severity score (MetS) for cardiometabolic health, and a lung function score (spirometry test) for respiratory health. Those ECRS are computed using recent non-parametric machine learning models (i.e., ensemble of trees), able to capture complex exposome-health relationships and interactions. Obtained ECRS explain a substantial portion of the variance, in particular for mental and cardiometabolic ECRS, and their performances generalize well across all six cohorts. We identify predictors with an overall high impact on the predicted risk, such as maternal stress, child BMI, and noise exposure for mental health. We also extract non-linear dose-response relationships. Our approach's main benefit lies in its ability to capture complex associations and extract insights at both a global and personal level for each exposure or groups of exposures. Overall, this study highlights the potential of such approaches to

compute risk scores able to inform practitioners about actionable factors of high-risk children.

## Methods

### Study participants

This study uses data from the HELIX project. This project includes data from six different European longitudinal birth cohorts, namely: (1) Born in Bradford alias BiB; UK[22], (2) Etude des Déterminants pré et postnatals du Développement et de la santé de l'Enfant, alias EDEN; France[23], (3) Infancia y Medio Ambiente alias INMA, Spain[24], (4) Kaunas Cohort alias KANC; Lithuania[25], (5) Norwegian Mother, Father and Child Cohort Study alias MoBa; Norway[26,27], and (6) Mother-Child Cohort in Crete alias RHEA; Greece[28]. Children were born at different periods depending on the cohort (RHEA 2007–2008, EDEN 2003–2005, INMA and MoBa 2005–2007 and KANC 2007–2009). In total nearly 32,000 mother-child pairs were initially followed during pregnancy and a subset into childhood from 6 to 12 years old depending on the cohort. From these, we used data from 1622 pairs for which biological samples, environmental exposures, clinical biomarkers and health outcomes were assessed with common standardized protocols. All six cohorts in which HELIX is based had undergone the required evaluation by national ethics committees (prior the start of the project) and confirmed that relevant informed consent was given for secondary use of the data[20].

### Data

**Health outcomes.** Health outcomes during childhood were either directly measured or derived from variables collected between December 2013 and 2016 in the Helix subcohort follow-up visit[20]. Hence, health outcomes and childhood environmental factors are cross-sectional.

**Mental health.** We modelled mental health using the P-factor, a reliable measure of psychopathology in youth populations[29,30] that represents life course vulnerability to psychiatric disorders[31] and is predictive of long-term psychiatric and functional outcomes[32]. P-factor in childhood has been found to predict the course and severity of a multitude of psychiatric outcomes in adolescence[33]. It was computed using confirmatory factor analysis (CFA) to fit a hierarchical general psychopathology model with the Lavaan statistical package[34] with data from the 99-item Child Behavior Checklist[35], a questionnaire filled out by the parents.

**Cardiometabolic health.** We used an aggregated metabolic syndrome score as a summary score for cardiometabolic health[36]. It was calculated using the z scores of waist circumference, systolic and diastolic blood pressures, levels of triglyceride, high-density lipoprotein cholesterol, and insulin with the following formula: metabolic syndrome = z waist circumference + (–z HDL cholesterol level + z triglyceride level)/2 + z insulin + (z systolic blood pressure + z diastolic blood pressure)/2. A higher metabolic syndrome score indicated a poorer metabolic profile.

**Respiratory health.** Finally, we assessed the lung function using the child forced expiratory (air) volume in one second (FEV$_1$) percent predicted value as in a previous HELIX study [37]. FEV$_1$ was measured with a spirometry test (EasyOne spirometer; NDD [New Diagnostic Design], Zurich, Switzerland) using a standard standardized protocol. Then the Global Lung Initiative reference equations[38] were used to compute FEV1 percent predicted values (i.e., values standardized by age, height, sex, and ethnicity of the patient).

**Environmental data.** We used a wide variety of environmental exposures from both mothers (during pregnancy) and children (between age 6 to 12 depending on the cohort of inclusion) that participated in the HELIX follow-up visit organized between December 2013 and 2016[20]. Measurements on pregnant mothers were collected between 1999 and 2010. Information about the methods used to estimate those exposures is available in Supplementary Notes, Supplementary Tables 1–9.

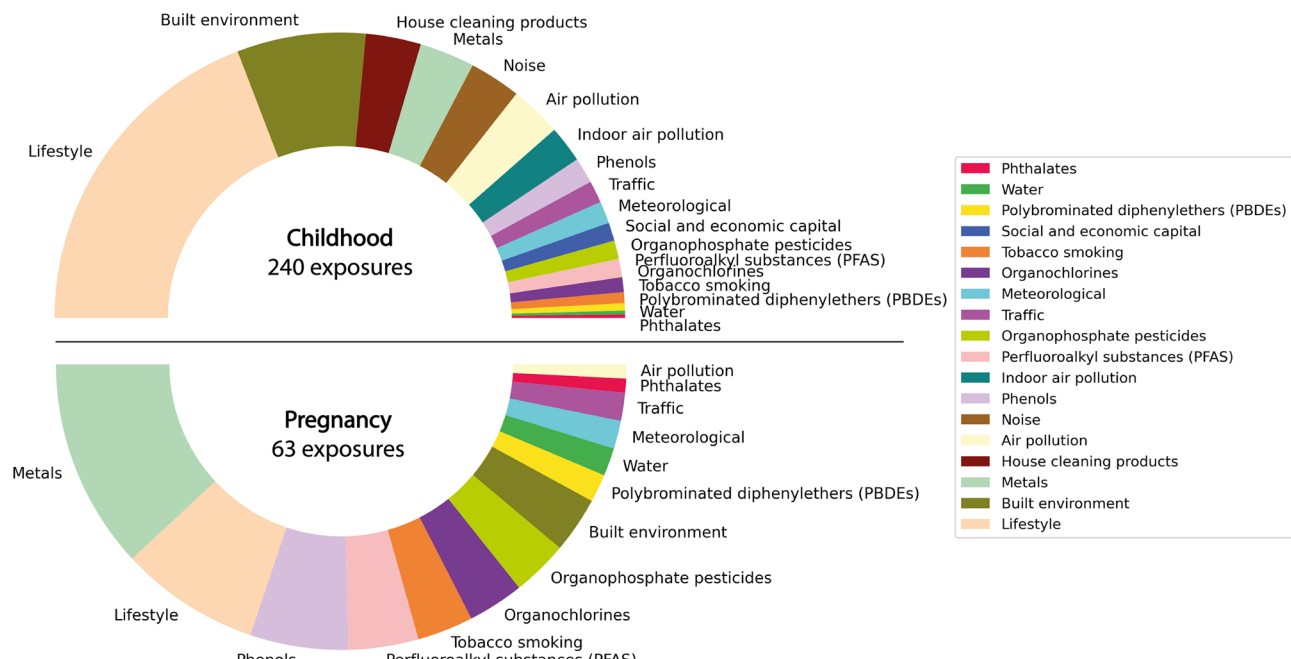

**Fig. 1 | Proportions of exposures grouped into families.** Pie chart displaying the 18 different families of exposures considered in the study and their relative sizes in terms of number of exposures considered within each family. The top half of the chart displays exposures measured during childhood ($n = 240$) while the bottom part displays those measured during pregnancy ($n = 63$).

In previous HELIX exposome wide studies (including chemical, outdoor, psychosocial exposures) a sub selection of variables was made among the questionnaire data, and they did not include together external and internal exposome. Due to the exploratory nature of our study, we included new variables previously unexplored in the HELIX studies. In total, we selected 63 prenatal and 240 postnatal exposures grouped in 18 exposure families. An overview of those families is given in Fig. 1. Most of the exposures used were already described in previous publications, specifically measurement available and baseline data[20]. New exposures, previously not described in detail, were extracted from the HELIX subcohort main questionnaire, more precisely about children's time spent outside (during weekends and holidays), noise disturbance, and house cleaning products. A full list and a description of the selected variables are available in Supplementary Data 1.

**Outdoor and indoor exposures**. Outdoor exposures were estimated using geographic information systems (GIS), remote sensing and spatiotemporal modeling[39]. Considered exposures include air pollutants (e.g., particulate matter), meteorological factors (temperature, humidity, UV exposure), traffic noise, traffic indicators, natural space (green spaces, blue spaces) and built environment (e.g., building density, public transport, facilities, etc.). More details are provided in Supplementary Notes, Supplementary Tables 1–4.

Indoor air pollution exposure to NO2 and to volatile organic compounds benzene, toluene, ethylbenzene, meta-xylene, para-xylene and ortho-xylene, was measured through passive samplers installed in the homes of 150 individuals in the panel studies and extrapolated for the whole HELIX subcohort using prediction models[20].

**Lifestyle**. Lifestyle exposures were collected using a standardized questionnaire developed for HELIX and included the child's diet, physical activity, sleeping patterns, socioeconomic variables (e.g., subjective wealth, social capital of the family), exposure to environmental tobacco smoke, water consumption habits, cleaning products, noise perception and time outdoors. Water disinfection by-product measurements were collected from water companies for the entire cohorts in each HELIX

center. More details are provided in Supplementary Notes, Supplementary Table 5.

**Biomonitored chemical pollutants**. Pollutant biomarkers were assessed during pregnancy and childhood using blood and urine samples. They include organophosphate pesticides, phenols, phthalates, metals, perfluoroalkyl (PFAS) substances, polybrominated diphenyl ethers (PBDEs), organochlorines and creatine. These measurements were adjusted for lipids and creatinine when appropriate. More details are provided in Supplementary Notes, Supplementary Tables 6–9.

Metabolites and proteins. We included 122 protein and metabolite measurements in the study. More specifically, we included 36 proteins that were assessed from plasma using Luminex immunoassay kits (cytokines 30-plex, apoliprotein 5-plex and adipokine 15-plex). Forty-two blood serum metabolite indicators were assessed using the targeted Biocrates' AbsoluteIDQ p180 kit and the MetIDQ™ RatioExplorer software that calculated sums and ratios of metabolites, termed metabolism indicators, to improve biological interpretation. Forty-four urine metabolites were assessed using proton nuclear magnetic resonance ($^1$H NMR) spectroscopy[40]. Urine metabolites were normalized using the median fold change normalization method[41], which takes into account the distribution of relative levels of all metabolites compared to the reference sample in determining the most probable dilution factor. The full list and the description of selected metabolites and proteins are available in Supplementary Data 1.

Parental and child clinical factors. Clinical factors were collected during childhood or pregnancy from the HELIX subcohort follow-up clinical examination (between December 2013 and 2016)[20] or initial cohort assessments on pregnant mothers (between 1999 and 2010). Childhood clinical factors included maternal mental and cognitive states (e.g., maternal perceived stress (short form version)[42], maternal working memory[43]), child respiratory factors (e.g. diagnosed asthma, self-reported rhinitis) and cardiometabolic factors (e.g., systolic and diastolic blood pressure, blood lipids). Pregnancy clinical factors only include maternal blood lipids collected

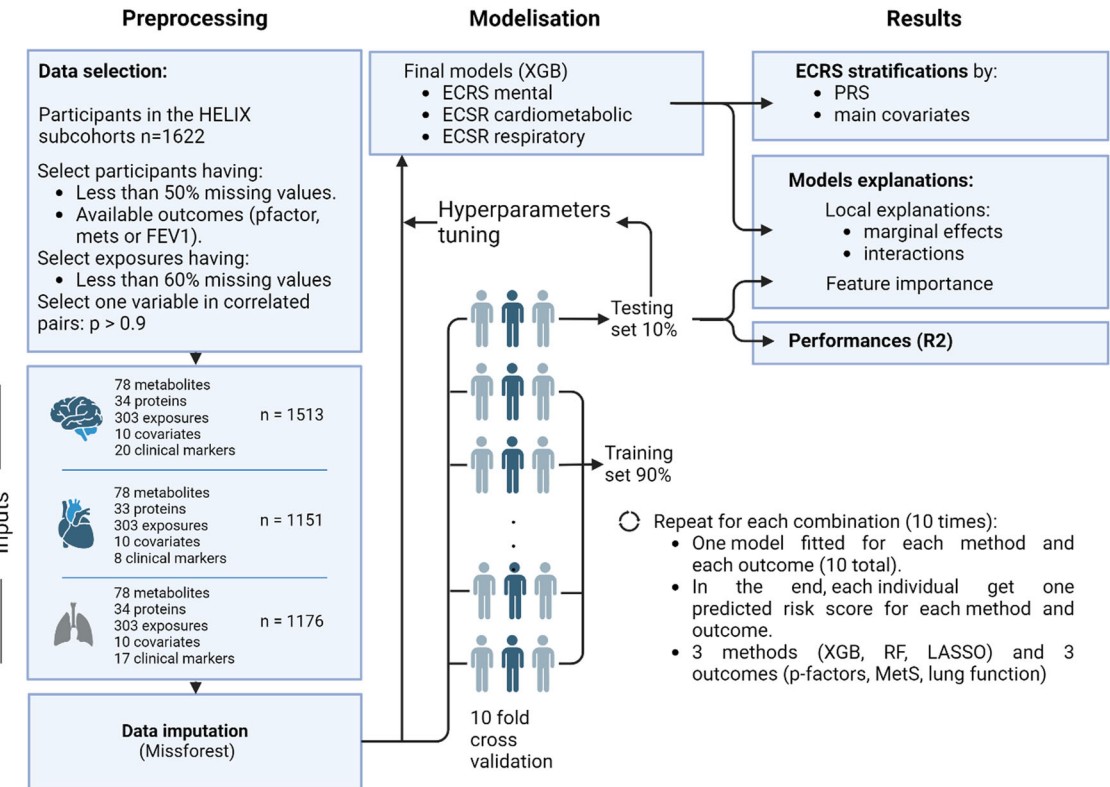

**Fig. 2 | Analysis workflow.** This figure provides a concise overview of the steps performed in the study analysis, organized sequentially within three main stages corresponding to three columns. These stages are the preprocessing of input data, the modeling of ECRS, and the reporting of results. Abbreviations: RF: Random Forest, XGB: XGBoost, ECRS: Environmental-Clinical Risk Score, MetS: metabolic syndrome, FEV1: Forced expiratory volume in 1 s (lung function).

during the initial cohorts' assessments. The complete list and the description of included clinical variables are available in Supplementary Data 1.

Covariates. Covariates were used as predictors in the ECRS. As covariates, we used both children (e.g., age at examination, sex, asthma medication, the season of birth) and parents' characteristics (e.g., parents' nativity, paternal and maternal education, mother age at birth, mother's parity). The full list and description are available in Supplementary Data 1.

### Statistical analysis
All data processing was performed in Python 3.9.7.

**Data preparation.** Figure 2 provides a brief description of the data selection process and the study workflow. This study aims to agnostically discover exposure-health associations while minimizing the likelihood of overfitting and maximizing ECRS models' interpretability. Hence, we performed several minimal data selection steps, from the initial selection of data to the filtering of strongly correlated and noisy data.

First, as detailed in the Method—Data section, we used a wide selection of previously described variables enriched with new exposures that were not assessed in previous HELIX studies. In this step, similarly to previous studies on the HELIX sub-cohort data[37,44], we selected single representatives from groups of related and correlated variables to reduce multicollinearity and increase interpretability. For instance, we only kept the 300 meter buffers for the number of road intersections per km² and removed the 100 meter buffers; or we considered only home-based air pollutants measurements and discarded those from schools or other places. The full description of the preselection steps is available in the annex (Supplementary Methods—part 1).

Then, we further refined this selection by filtering among strongly correlated variables ($r > 0.9$), discarding 28 variables in total. The rules

applied for this step are also described in the annex (Supplementary Methods—part 2).

To reduce the amount of missingness in the data, we discarded records of individuals with >50% (102 records discarded) and variables with >60% missing data (3 variables discarded). More information about each selection step is provided in Supplementary Table 10. The mean percentage of missing values per exposure was 14% (first quartile 0.66% and third quartile 20.59%). Percentages of imputed missing values for each selected variable are available in Supplementary Data 1 and computed for each cohort in Supplementary Data 2. Missing values were imputed once using MissForest[45], a single iterative imputation algorithm that can handle both categorical and continuous variables and capture nonlinear relationships. We compared the performance of this method with two classical single imputation methods, namely KNN and mean imputation on manually generated missing values. The number of missing values to add on top of the originals was settled to be proportional to 15% of the total sample size for each variable (i.e., 243). MissForest considerably outperformed KNN and mean imputation with a mean squared error of 0.56, 1.00 and 1.25 respectively. From this imputed dataset, individuals with non-missing outcomes were selected for the mental, cardiometabolic, and respiratory risk scores resulting in three distinct datasets.

Finally, depending on the outcome for each dataset, we excluded clinical factors closely related to the outcome to prevent data leakage, which could over-inflate the model's predictive power (e.g., blood pressure for MetS). The selection made on the basis of the outcome is available in annex (Supplementary Data 1).

After selection, the data were prepared for the analysis. Depending on their nature, categorical data were one hot encoded (i.e., changed into dummy variables), labeled, or encoded with floating values (for frequencies, binned continuous variables, etc.). Out of the 478 total selected variables, 75 were categorical (11 one-hot encoded, 43 labeled). Then each phenotype was standardized into z-scores.

**Modeling.** We computed ECRS with supervised machine learning methods, predicting simple scalar measures as outcomes (P-factor, MetS and lung function) and using multiple environmental and clinical variables, including metabolites and proteins as predictors. Training is first performed in a 10-fold cross-validation (CV) procedure, where hyperparameters of the methods are optimized and performances measured. A 10-fold CV was chosen over, for instance, a leave-one-out or a 5-fold CV to balance the computational efficiency and the robustness of our performance estimates. Hyperparameters optimization of each method was achieved using the Tree-structured Parzen Estimator[46] from the Optuna library, which optimizes the path through the hyperparameters space. Supplementary Tables 11 and 12 show the selected hyperparameters for each model.

We compared predictive performances of ECRS computed with three methods:

The first method is LASSO, a penalized linear regression method widely used in the field. Note that technically, while the model itself is linear (it models the relationships between the input features and the output using a linear equation), the optimization process due to the L1 penalty is nonlinear. One of its main advantages is its ability to handle high-dimensional data through regularization. For this method, data was standardized before training.

The second and third methods are nonlinear and non-parametric ensembles of trees (i.e., Random Forests and eXtreme Gradient Boosting aka XGBoost[47]). Ensemble methods are able to handle small datasets and high dimensionality[48] while interaction effects can be captured and extracted from tree models[49,50]. In terms of prediction power, tree-based methods are still competitive with Deep Neural Networks (DNN) on tabular data[51] due to (1) their robustness to uninformative features, (2) their ability to preserve the data orientation, and (3) their ability to easily learn irregular functions. Machine learning, with systematic out-of-sample testing, employs a rigorous approach to test how well relationships between targets and variables are captured. The better a model performs, the more accurate associations it captures, and very poor performances may indicate unreliably captured information.

One of the HELIX project particularities is that it aggregates data from six different cohorts, with some variables having very distinct distributions across them[52]. Those variables are likely to be biased by cohort-related effects. Thus, we chose to penalize contributions of features strongly associated with the cohorts proportionally to their importance in the prediction. Each ECRS was computed using the following modeling (including those in the CV procedures):

$$Y = f_0(X) + g_0(Z) + R,\ E[R|Z, X] = 0$$

Where Y is the phenotype, X the environmental/clinical factors and usual covariates, Z the one hot encoded cohorts of inclusion, and R the residuals. f0 and g0 were estimated using our regressive methods (LASSO, RF, XGB) in two sequential steps with separate models.

$$\text{Step1}: Y = g_0(Z) + U,\ E[U|Z] = 0 \tag{2}$$

$$\text{Step2}: U = f_0(X) + V,\ E[V|X] = 0 \tag{3}$$

**Models' explanations.** Unlike LASSO, tree-based approaches can capture complex relationships that are not limited to a single coefficient per feature. We used SHAP[53], a local explanation method that uses Shapley values to extract different contribution coefficients for each individual. This allowed us to keep nonlinear relationships in the explanations. Shapley values were initially used in cooperative game theory to estimate the contribution of each player to the overall cooperation with desirable properties[54]. Adapted to machine learning, it gives the contribution of each feature to the overall prediction at the local (individual) level and can be aggregated (taking the average absolute Shapley values) to give a global measure of feature importance.

Concretely, a Shapley value gives, for a given individual, how much the given associated variable impacted the model prediction from the mean predicted value (negatively or positively). They are additives and sum up to the mean predicted value. We used them to explore captured associations (e.g., their directions, nonlinearities) and to compute measures of the global importance of a feature (or a family/group of features) obtained by averaging its absolute (Shapley) values across individuals. Because Shapley values are additives, the contribution of a group of features is easily computed (taking the sum of Shapley values at the individual level). We also computed SHAP interactions[50] to explore potential (pairwise) cocktail effects.

Measures of global feature importance were computed in the 10-fold CV loop to estimate confidence intervals. Features importances were computed using XGBoost, the best performing nonlinear tree-based method. Local explanations and stratification were conducted on final ECRS obtained by training the XGBoost models on the whole dataset, with the hyperparameters selected by the 10-fold CV procedure. We also computed ECRS with Lasso on the whole dataset to compare extracted Shapley values.

**Sensitivity analysis.** Finally, for each ECRS, we tested the robustness of our method when applied to different populations. Leveraging the six different cohorts' data available in our study, we applied a leave-one-cohort-out cross-validation procedure, recursively training our XGBoost models on five cohorts to predict the sixth. Before training, we standardized both features and targets (e.g., P-factor, MetS and lung function) across each cohort. The hyperparameters used were the same as before and were not optimized for this task.

## Ethics approval
Local ethical committees approved the studies that were conducted according to the guidelines laid down in the Declaration of Helsinki. The ethical committees for each cohort were the following: BIB, Bradford Teaching Hospitals NHS Foundation Trust; EDEN, Agence nationale de sécurité du médicament et des produits de santé; INMA, Comité Ético de Inverticación Clínica Parc de Salut MAR; KANC, LIETUVOS BIOETIKOS KOMITETAS; MoBa, Regional komité for medisinsk og helsefaglig forskningsetikk; Rhea, Ethical committee of the general university hospital of Heraklion, Crete. Informed consent was obtained from a parent and/or legal guardian of all participants in the study.

## Reporting summary
Further information on research design is available in the Nature Portfolio Reporting Summary linked to this article.

## Results
### Population characteristics
To investigate the predictive potential of early-life external and internal exposome associated with mental, cardiometabolic, and lung health in children, we selected a study cohort of 1622 mother-child pairs who participated in the HELIX study. This cohort was composed of approximately half females (46.1%), mainly of European ancestry (82.9%), from highly educated families (40.1% with high maternal education), and the majority residing in urban areas (75.3% in areas with a density of population > 1500 inhabitants/km²) (see Supplementary Fig. 1). At the time of the health assessment, children were on average 8 years old (range: 5.5–12 years), 3.9% regularly visited the psychologist, and 7.3% had a neuropsychiatric diagnosis at the time of the visit (according to parent's reports, besides the CBCL screening). Based on the World Health Organization (WHO) international standards for BMI cut-offs (normal: 18.5–25 kg m-2, overweight: 25–30 kg m-2, obese: ≥ 30 kg m-2), while 69.2% of participants fell within the normal category (n = 1122), 10.6% of participants were categorized as overweight (n = 172), and 20.2% of participants were identified as obese (n = 328). Additionally, 10.2% of children were reported to have asthma (ever diagnosed).

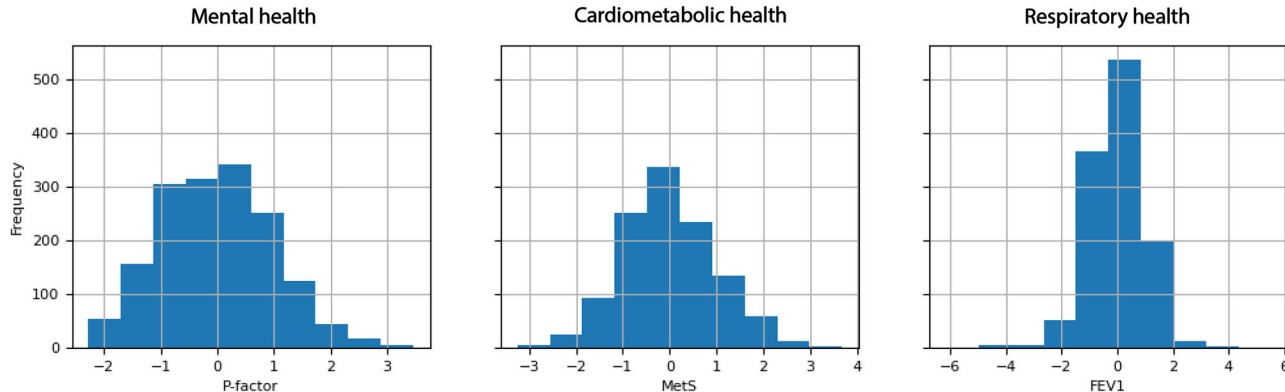

**Fig. 3 | Standardized health outcome distributions measured in 6–12 year-old HELIX children.** Each histogram represents the distribution of a health outcome grouped into 10 bins. The horizontal axis shows the range of the outcome, and the vertical axis shows the number of children falling within each bin. Abbreviations: MetS: Metabolic Syndrome, FEV1: Forced Expiratory air Volume in 1 s.

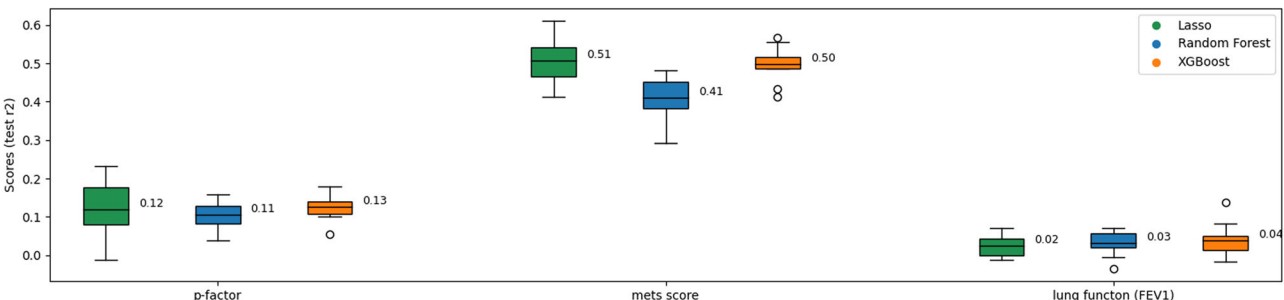

**Fig. 4 | Models' performance comparison obtained after cohort adjustment.** The box extends from the first quartile (Q1) to the third quartile (Q3) of the data computed from the ten models ($n = 10$) in the cross-validation procedure, with a line at the mean. The whiskers extend from the box to the farthest data point lying within 1.5 x the inter-quartile range (IQR) from the box.

Leveraging data from the pregnancy ($p = 62$) and childhood ($p = 241$) exposome, preclinical biomarkers ($p = 112$), and clinical factors ($p = 18$) along with covariates ($p = 14$) (full list available in Supplementary Data 1), we trained machine learning models to predict the *P*-factor, MetS, and lung function. We used 445 features from 1513 individuals for mental health, 432 from 1151 for cardiometabolic health, and 442 from 1176 for respiratory health (Fig. 2). P-factor and metabolic syndrome were square root transformed to be normally distributed. All three health outcomes were standardized to have a mean of 0 and a standard deviation of 1, with ranges of −2.27–3.46, −3.25–3.68, and −4.80–5.60 for mental, cardiometabolic, and respiratory health, respectively (Fig. 3). After transformation and standardization, all outcomes were normally distributed with two-sided Kolmogorov-Smirnov test *p*-values of 0.08, 0.11 and 0.24 for *P*-factor, MetS and lung function respectively. For *P*-factor and MetS, higher scores indicate an increased risk and for lung function a decreased risk.

### Predictive performances: environmental-clinical features captured 13%, 50%, and 4% of the variance in mental, cardiovascular, and respiratory health

To address overfitting, improve stability, and compare model performances and generalizability for all children, we implemented a ten-fold iteration scheme for tested algorithms (XGBoost, RF, and LASSO) with cross-validation (CV) (Fig. 2; see Method). This approach generated, for each algorithm, ten fitted sparse models for each outcome. The comparative analysis of all methods' predictive performances is presented in Fig. 4. These performances were obtained after cohort adjustment (see Method, part 2: Modeling). Cohorts accounted for 5 to 14% of the out-of-sample variance (see Supplementary Fig. 2).

LASSO models explained around 12% of the variance in the P-factor, 51% in MetS, and 2% in lung function. In contrast, RF explained 11% of the variance in the P-factor, 41% in MetS, and 3% in lung function. Finally, XGBoost explained 13% of the variance in the P-factor, 50% in MetS, and 4% in lung function (Fig. 4). Across the three outcomes, the differences between LASSO and XGBoost were not statistically significant, as confirmed by 10-fold cross-validated two-sided paired student *t*-tests with *p*-values of 0.686, 0.656, and 0.216 for P-factor, MetS, and lung function, respectively. Information about the residuals obtained during the cross-validation procedure is summarized in Supplementary Table 13. They were, on average, centered around 0 and normally distributed across all CV folds unless for the respiratory health scores that were considered normal less consistently.

### Global feature importance: ECRS captures the influence on health of both external and internal exposome factors

From the three ECRS computed with XGBoost, the best performing nonlinear method, we computed Shapley values for each feature and each individual using SHAP. Averaging the absolute values of Shapley values across all individuals gives a measure of the global impact of each variable (or groups of variables) on health outcomes. Fig. 5 shows the feature importance for the top 20 variables at the level of each feature and of exposure families for all phenotypes obtained from the 10 fitted cross validated XGBoost models. More exhaustive feature importance lists (top 100) are available in Supplementary Data 3 for mental, cardiometabolic and respiratory ERS. In addition, the overall importance of all metabolites and proteins compared to exposomic variables, clinical factors and covariates is displayed, for each ECRS, in Supplementary Fig. 3.

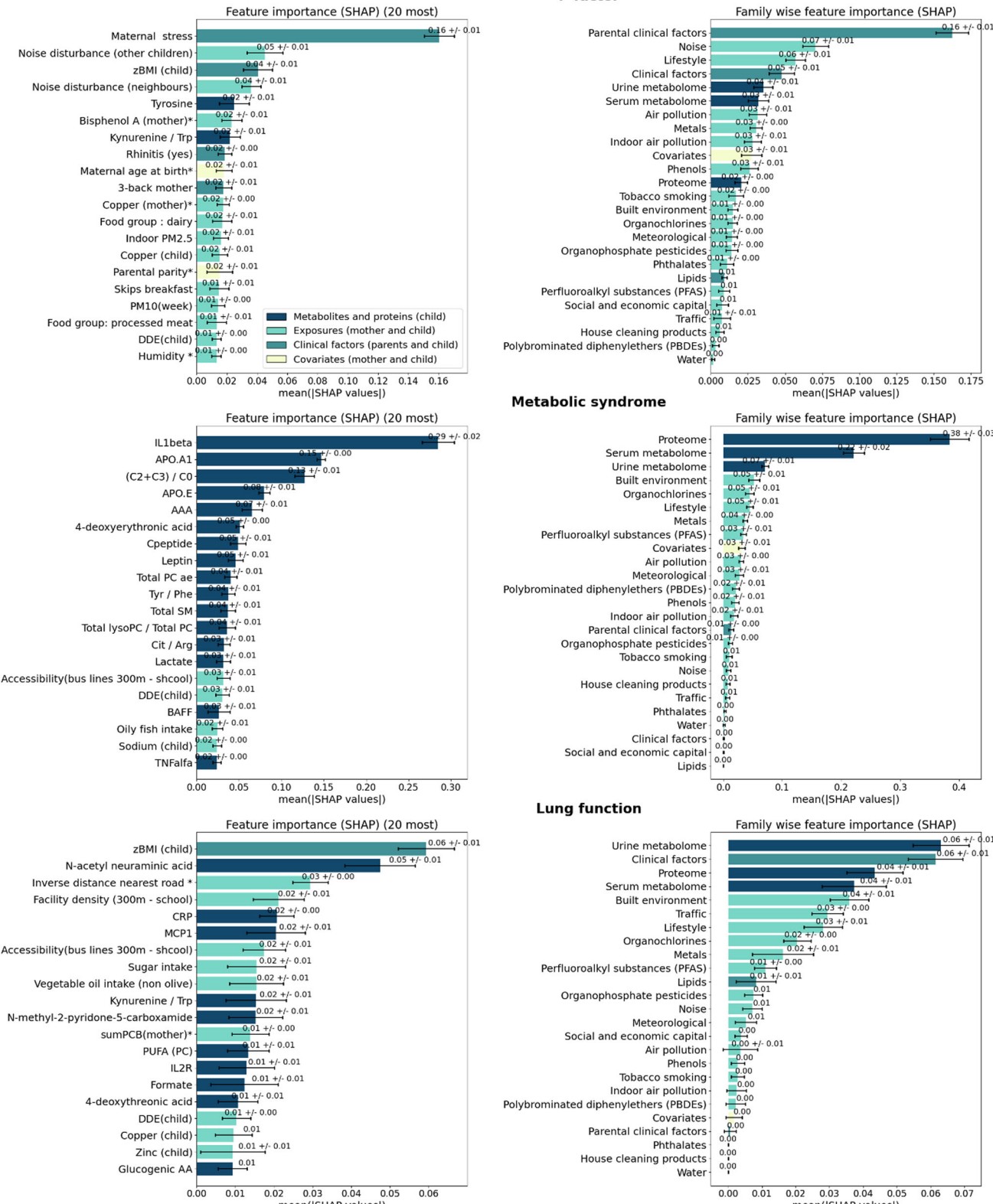

**Fig. 5 | Global feature contributions to the three environmental-clinical risk scores in the HELIX mother-child pairs.** Mean contributions are estimated from Shapley values for each individual factor (left column) and each family of factors (right column). The black interval bars represent the standard deviation across the ten models ($n = 10$). Only the top 20 most impactful factors are displayed here. Extended lists of feature contributions for each ERS are available in Supplementary Data 3. Variables assessed during pregnancy are indicated by an *.

For the P-factor, maternal stress was by large the most important feature, with a mean SHAP of 0.16, followed by noise disturbance from other children with a mean of 0.05 and zBMI with a mean of 0.04. Apart from parental clinical factors with a mean SHAP of 0.16 (mostly driven by the impact of maternal stress), noise disturbance, and lifestyle exposures (such as skipping breakfast, dairy intake and processed meat consumption) were the most important families of exposures, with mean of 0.07 and 0.06, respectively. Other factors not belonging to these families such as tyrosine (urine metabolite) with a mean SHAP of 0.02 and prenatal bisphenol A (phenols) with a mean SHAP of 0.02 were also noteworthy.

For MetS, the interleukin-1 beta (IL1B) protein was the most prominent feature with a mean SHAP of 0.29. Proteins, serum and urine metabolites, as families of variables, exhibited the most impact on the predicted phenotype. They were largely driven by IL1B, Apolipoprotein A1 (APOA1), and the ratio of short chain acylcarnitines to free carnitine ((C2 + C3)/C0). Overall, for this ECRS, metabolites and proteins combined had a mean SHAP of 0.50 while exposures had 0.14 and clinical factors 0.03 (Supplementary Fig. 3).

Finally, for lung function, although the XGBoost model could explain only a minor part of the outcome (4%) and therefore warrants precaution in the interpretation of the feature importance, no individual features were standing out compared to the other health outcomes. The most important features were child zBMI, *N*-acetylneuraminic acid (Neu5Ac) and the inverse distance to the nearest road during pregnancy (Fig. 5).

On average, childhood measurements were 3.03–8.95 times more important than prenatal variables for all risk scores. We found that childhood factors had a mean contribution of 0.23 (standard deviation: 0.01), in comparison with a prenatal mean contribution of 0.06 (standard deviation: 0.01). For cardiometabolic health, the postnatal mean contribution was 0.51 (standard deviation: 0.03) while prenatal factors had a mean contribution of only 0.06 (standard deviation: 0.00). For respiratory health, the postnatal mean contribution was 0.13 (standard deviation: 0.01), while the prenatal mean contribution was only 0.04 (standard deviation: 0.01).

## Local feature importance: Local explanations reveal complex nonlinear relationships between exposures and health outcomes

Unlike linear models where feature coefficients are identical for all individuals, SHAP extracts contributions that are specific to each individual, allowing to assess more complex exposome-health relationships than regression coefficients. The ECRS captured both linear and nonlinear relationships, as shown by SHAP dependence plots (Fig. 6 and Supplementary Fig. 4). For instance, the relationship between maternal stress and noise disturbance from neighbors followed a linear trajectory, while the impact of child zBMI on the P-factor displayed a more complex pattern with a threshold effect. These plots also allowed us to visualize the directions of the distinct associations for each outcome.

A high value in maternal stress was related to an increase in the predicted P-factor, indicating an increased risk for mental health issues. Low levels of noise disturbances and child zBMI slightly reduced the risk of mental health problems, while high values had a particular harmful impact, especially in the case of child zBMI.

For cardiometabolic health, IL1B was positively associated with MetS, with high values having an important impact on the risk. We observed similar relationships, to a lesser magnitude for AAA, apolipoprotein-E (APOE) and C-peptide. Conversely, high values of APOA1 and the (C2 + C3)/C0 ratio showed a strong protective impact.

The results for the respiratory health scores should be interpreted with caution because of the low variance explained by the model after cohort adjustment (4%). We observed a protective impact on lung function for child BMI and inverse distance from the nearest road during pregnancy. Low BMI values substantially increased the risk, while moderate to high values slightly reduced it. On the contrary, high values of Neu5Ac, facility density near schools (300 m), CRP, Monocyte Chemoattractant Protein-1 (MCP1), sugar and oil intake were associated with decreased lung function.

Compared with the linear associations obtained with Lasso (Supplementary Fig. 5), directions of associations were consistent with XGBoost, with some exceptions (e.g., leptin for cardiometabolic health and bus line accessibility for respiratory health). Overall, predictions obtained with XGBoost were more conservative for extreme values of the predictors (e.g., maternal stress for mental health or child zBMI for respiratory health, etc.).

## Pairwise input interaction effects were extracted from captured relationships

Pairwise interaction effects were derived from Shapley values using SHAP. Supplementary Figs. 6 and 7 show plots for the top ten interactions (according to the mean absolute value of Shapley values) derived from the mental and cardiometabolic risk scores. For lung function, the predictive power of the risk score was insufficient to extract meaningful information from its captured interactions. Overall, interaction effects on predicted risk were relatively small compared to the marginal effects. We observed a 7.4–8.8 ratio between the mean top ten marginal effects and the mean top ten interaction effects, depending on the outcome of interest (respectively, 0.042 and 0.005 for mental health, and 0.093 and 0.013 for cardiometabolic health). This indicates that overall, pairwise interactions had a much smaller impact than marginal relationships on the predicted risk for the two scores.

For mental health, the most important captured interactions were between perceived maternal stress during follow-up assessment and factors from diverse exposure families (clinical factors, lifestyle, noise disturbance, etc.). Specifically, the most important interactions were between the following factors: maternal stress and allergic rhinitis, with a mean SHAP value of 0.0070; maternal stress and insulin, with a mean SHAP value of 0.006 and maternal stress with dairy intake, with a mean of 0.006 (Supplementary Fig. 6). For cardiometabolic health, the top ten most relevant captured interactions were between clinical biomarkers (proteins and metabolites) or between IL1B and other factors, such as temperature or child's age. Specifically, the most important captured interactions were between the following factors: IL1B and ratio of short chain acylcarnitines to free carnitine, with a mean SHAP value of 0.025; APOA1 and APOE, with a mean of 0.016 and APOA1 and the ratio of short chain acylcarnitines to free carnitine with a mean of 0.013 (Supplementary Fig. 7).

Figure 7 shows two arbitrary interactions, selected from the top ten most important ones, derived from the mental and the cardiometabolic risk scores. The first interaction is between maternal stress and the insulin measured in children and impacts mental health. It indicates a harmful impact of high insulin combined with maternal stress. The second interaction for cardiometabolic health is between IL1B and the ratio of short chain acylcarnitines to free carnitine. It indicates a significative impact of this ratio on children with high values of IL1beta, with a high ratio value associated with higher risk and vice-versa.

## Overall ECRS performances generalize well on different populations with discrepancies across cohorts

For each of the three ECRS, performances obtained from the leave-one-cohort-out cross-validation were consistent with those obtained using the ten-fold cross-validation. The explained variance was 13.4% (4% std), 46.8% (12% std) and 2.4% (2% std) for mental, cardiometabolic and respiratory health respectively. Differences in predictive performances were observed for all ERCS depending on the left-out cohort. For instance, the ECRS for the P-Factor was less predictive of the outcome when the KANC cohort was predicted based on all the other cohorts ($R^2$ = 6.3% versus 13.4 on average). Table 1 shows the variance explained within each cohort.

## Discussion

This is the first study that computed children's ECRS for mental, cardiometabolic, and respiratory health, and covered a wide range of pre- and post-natal factors (including air pollution, noise, urban and social environment, lifestyle, chemical exposures, metabolites, proteins, and clinical factors). The inclusion of data from six different cohorts across different countries allowed us, by adjusting our models for cohorts, to extract non

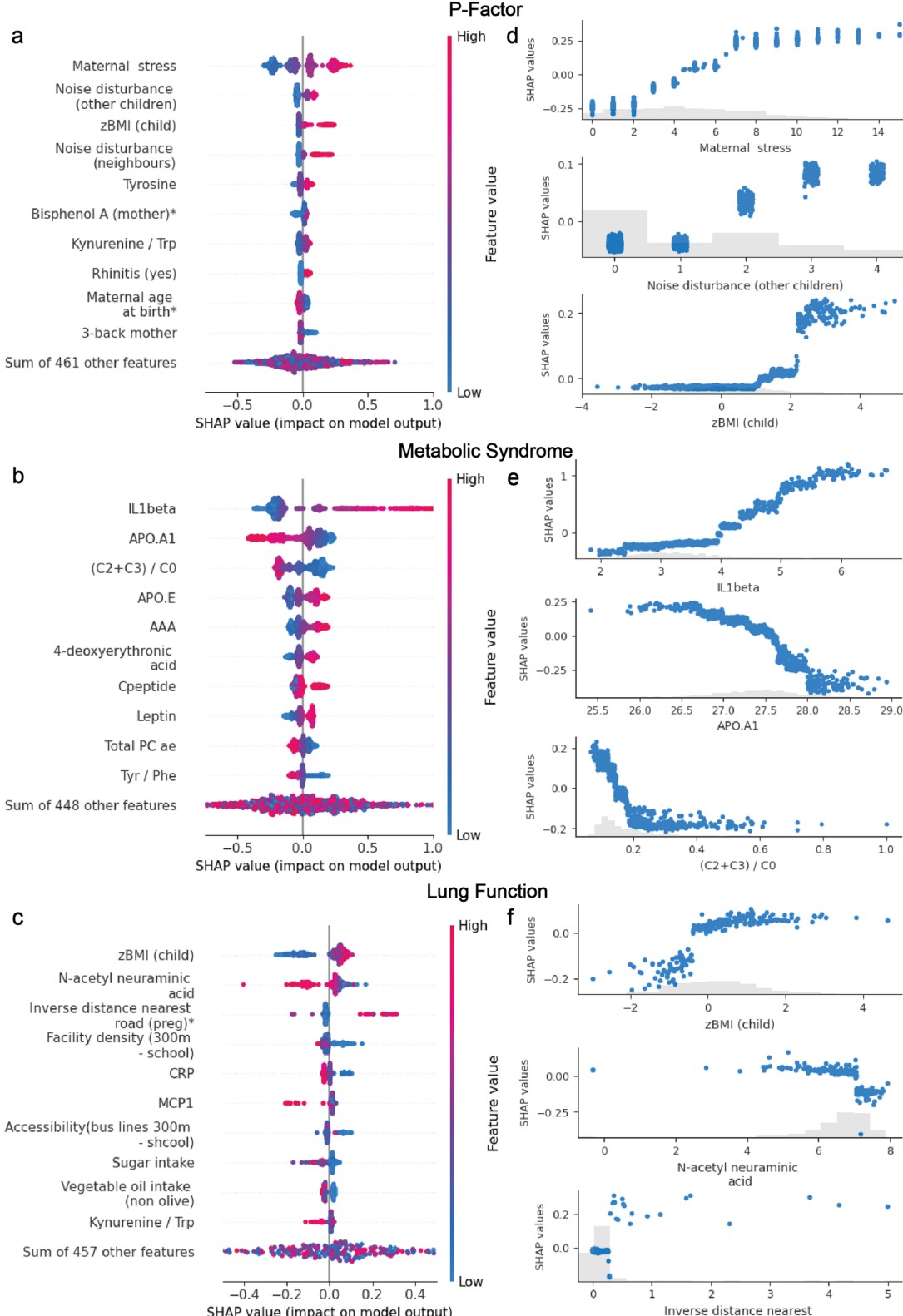

**Fig. 6 | Local explanations (SHAP) from the three environmental-clinical risk scores in HELIX mother-child pairs. a–c** Beeswarm plots of Shapley values for the ten most important features for mental, cardiovascular and respiratory ECRS respectively. Each dot represents the contribution (Shapley value) of a feature for a given individual in the model's prediction. Dots accumulate along each feature to show density. The feature value for each individual is shown in a colored range from low to high. **d–f** Dependence plots of top three most important features for mental, cardiovascular and respiratory ECRS respectively. Each dot represents the contribution (Shapley value), on the *y*-axis, of a feature, on the *x*-axis, for a given individual in the model's prediction. Gray bars show the features' distributions. Variables that were assessed during pregnancy are indicated by an *.

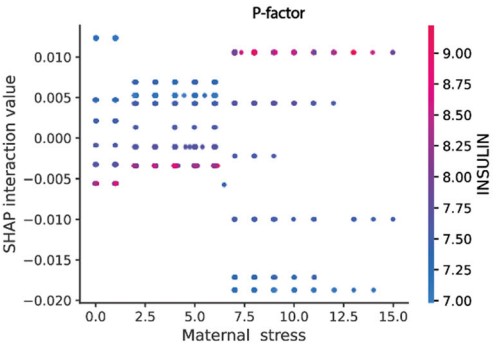
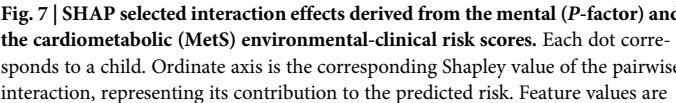
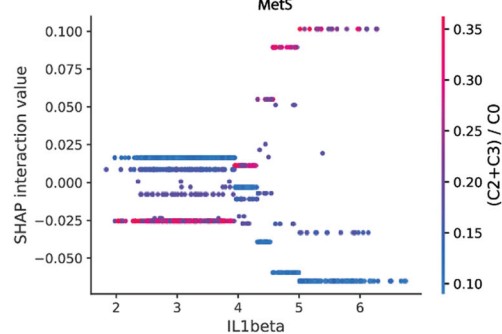

**Fig. 7 | SHAP selected interaction effects derived from the mental (*P*-factor) and the cardiometabolic (MetS) environmental-clinical risk scores.** Each dot corresponds to a child. Ordinate axis is the corresponding Shapley value of the pairwise interaction, representing its contribution to the predicted risk. Feature values are given on the *x*-axis (first feature) and on the colored scale (second feature). Marginals and interactions effects are additive for each individual and sum to the predicted value.

**Table 1 | Variance explained ($R^2$) by each ECRS in the leave-one-cohort-out cross validation procedure**

| | Cohorts | | | | | | | |
|---|---|---|---|---|---|---|---|---|
| | **BiB** | **EDEN** | **KANC** | **MoBa** | **RHEA** | **INMA** | **Mean** | **Standard deviation** |
| P-Factor | 16.9 | 11.8 | 6.3 | 15.5 | 13.3 | 16.7 | 13.4 | 4 |
| MetS | 53.2 | 47.3 | 41.6 | 59.4 | 56.4 | 22.7 | 46.8 | 12 |
| Lung function | 3.4 | 3.6 | 2.8 | 1.0 | −0.8 | 4.6 | 2.4 | 2 |

Our XGBoost based machine learning procedure were used to predict each cohort while trained on the others.

cohort-specific relationships, thereby increasing their likelihood of being generalizable. Predictive performances were superior for the cardiometabolic risk score ($R^2$: 50%), in particular driven by the (pre)clinical biomarkers compared to the other two health domains, $R^2$: 13% for general mental health and 4% for lung function. The most important variables were the following: parental clinical factors (mainly maternal stress), noise disturbance (mainly from neighbors and other children) and lifestyle exposure for mental health; protein and metabolites (mainly IL1B) for cardiometabolic health; and child BMI and urine metabolites for respiratory health. While our results need to be validated on an external population (e.g., different from the countries available in HELIX), the cohorts-based sensitivity analysis (leave-one-out cross-validation) showed promising results regarding the generalizability of our ECRS on European populations. Our approach's main benefit lies in its ability to capture complex associations and extract insights at both a global and personal level for each exposure or groups of exposures. Results showed several important captured relationships that were nonlinear.

Our study was performed in a context where current clinical tests often fail to identify children at risk of developing chronic diseases, notably regarding mental and cardiovascular diseases, which is a key challenge to the development of effective prevention and treatment policies. This limitation is largely attributed to the fact that many of these tests were developed and validated primarily in adult populations and to the paucity of longitudinal data on children. In the case of cardiovascular risk, the role of adipokines is a pertinent example. While the association between elevated inflammatory biomarkers and increased cardiovascular risk is well established in adults, there is a lack of comprehensive studies on the onset and progression of these biomarkers in children[55]. Therefore, more prospective studies focused on children are necessary to provide guidance to the pediatric medical community for more effective interventions and prevention policies.

Polygenic risk scores have substantially improved predictions in comparison with single genetic factors and its potentials for screening, prevention, treatment and disease management start to become apparent suggesting that their environmental analogue will be equally valuable in public health prevention[18], for both the identification of individuals and environmental factors of risk. A recent study from the UK Biobank[56] showed

that, in addition to standard clinical risk factors such as sex, age, blood pressure or BMI, ERS provides a greater increase in predictive performances compared with PRS. Furthermore, ERS captures holistic individual-level non-hereditary risk associations, providing clinicians with actionable factors of high-risk patients that are independent of genetics and provide guidance for prevention and treatment. In this case the environmental factors include, among others: noise disturbance, prenatal bisphenol A and ambient humidity for mental health; bus lines accessibility, child's dichlorodiphenyldichloroethylene and oily fish intake for cardiometabolic health; facility density, bus lines accessibility and sugar intake for respiratory health.

A known limitation of most nonlinear machine learning approaches is the model identifiability which can lead to complications such as overfitting or ambiguity in interpreting the model's parameters[57,58]. However, we believe our approach has several strengths that mitigate these concerns. First, our modeling strategy, backed by rigorous cross-validation, limits overfitting and prioritizes generalizability across different data subsets. Second, both nonlinear methods used in this study aggregate results from multiple models, whether from boosting or bagging, which enhances the stability of results. Finally, it is worth remembering that identifiable models such as linear regression or LASSO are also limited. They cannot capture nonlinear relationships and their interpretation are correct only if all relationships are linear.

Our tree-based approach has shown comparable predictive performances to LASSO, which could indicate that the relationships to capture are mostly linear in nature and that there are no interactions to be captured. However, the difference in prediction is likely to be due to the ability of simpler models to perform better in situations of small training sample size[57]. More data would be needed in order to confirm the nature of those relationships and, even if LASSO performs well, it may not capture all the complexities within the data, especially nonlinear relationships and interactions, which other algorithms could reveal. In this study, we explored such relationships using, to our knowledge, a novel approach in this field. We did not aim to assess nor confirm the causality of extracted relationships, but rather to identify new associations that previous methods would have missed or refine the potential nature of known ones (e.g., nonlinearities,

interactions). Thus, the causality of those findings needs to be validated in a causal inference framework.

We compared our findings to those previously obtained in the literature and highlighted novel relationships. Besides nonlinearities that are rarely assessed in the field, we found our results, when comparing only directions of associations, to be mainly consistent with those observed in other studies, which indirectly validates our approach.

For mental health, previous studies have found that maternal stress can lead to an increased risk of anxiety, depression, and behavioral problems in children[59], while a higher BMI has been associated with depression, anxiety, and low self-esteem[60]. Similarly, exposure to noise has been linked to both internalizing and externalizing behavioral problems in children[61]. Overall, while the majority of factors identified in this study have been linked with mental health outcomes in the literature, the exact causal pathways, the potential interactions between these factors, and the nuances of their impacts during specific developmental windows would benefit from further investigation. This is special true for the following environmental factors that are rarely studied in clinical mental health research: perceived noise disturbance, pollutant exposure to bisphenol A (endocrine disruptor chemical), and certain metabolic markers such as tyrosine levels and the Kynurenine/Tryptophan ratio.

For cardiometabolic health, several of the identified markers are well established (e.g., IL1beta[62], APOA1[63], Leptin[64]) and their known relationships are consistent with our findings. Other markers, such as aromatic amino acids or plasmalogens have been linked to several aspects of metabolic and cardiovascular health[65–67], but their causal pathways are still areas of ongoing research. Finally, 4-deoxyerythronic acid is not well covered in the literature.

At last, concerning respiratory health, we mainly identified a positive (nonlinear) association between FEV1 and child BMI, which supports the obesity paradox in chronic obstructive disease[68,69]. Unexpectedly, the inverse distance to the nearest road during pregnancy was associated with an increased FEV1. This association was already reported in a previous HELIX study[37] and is driven by the RHEA cohort. Overall, while our study identified relationships already well established (e.g., air quality through facility density or accessibility), others might be less direct and require further investigations (e.g., N-acetyl neuraminic acid, sugar and vegetable oil intake).

This study benefited from the richness of the HELIX project data, which used standardized outcomes, clinical biomarkers, and exposure measurement methods across six different European countries. The HELIX project used a wide range of exposure measurement techniques to collect both internal and external exposome data. In contrast to previous studies where ERS are usually derived from weighted sums of a limited number of exposures, our approach simultaneously assessed the impact of a wide range of exposures, metabolites, and clinical factors on several health outcomes. To address multicollinearities and nonlinearities, we used penalization and recent AI modeling techniques. SHAP allowed us to decompose the complex relationships captured by our ECRS for each feature, at a global and personal level, extracting interactions and marginal effects. Our risk scores were adjusted for the cohorts of inclusion, which is a particularity of the HELIX project. We acknowledge that in the case of highly correlated variables, our approach exposes those with the estimated higher impact on the predicted risk without implying causality. Correlations between exposures are known to present a challenge for exposome research, especially in the ability to differentiate true predictors from correlated covariates[70].

The main limitation of our study is the lack of external validation using an independent cohort. While our study benefited from the richness of exposome data not found in any other early-life cohort studies or preclinical studies, our sample size was relatively small ($n \sim 1500$), especially for applying certain machine learning methods (e.g., deep neural networks). This sample size limited our ability to capture and extract complex relationships, such as interaction effects, and favored our choice towards tree-based ensemble machine learning methods. Nevertheless, we observed a similar predictive value with LASSO. The other outcomes were both precomputed composite risk scores (P-factor and MetS), possibly suggesting better performances of machine learning methods in predicting raw outcomes. Additionally, as our study is exploratory in nature, we did not assess the causality of extracted exposome-health relationships. Further work is required to validate these relationships using a proper causal inference methodology. For instance, our scores might capture bidirectional cause-and-effect relationships, such as, potentially, the association between maternal stress and child behavior. The inclusion of internal peripheral markers that reflect the body's response to the measured health outcomes (e.g., IL1B and obesity) could also reinforce this phenomenon. In the case of MetS, the array of proteins and blood metabolites primarily covered biological pathways related to cardiometabolic outcomes, such as blood lipids. By decomposing the importance attributable to metabolites/proteins and the other factors (Supplementary Fig. 3), and further refined into families of factors, we limit this issue. The integration of these preclinical biomarkers aimed to enhance the predictive power of our ECRS and to investigate their relationships and interactions when combined with a wide variety of factors.

Another limitation is that our study does not account for the risk evolution over time since we used pregnancy and cross-sectional epidemiological data to assess childhood exposures at a single point in time, without considering their impact throughout an individual's lifetime. This limitation will be addressed in ATHLETE, the HELIX follow-up project in which the same children were regularly monitored into adolescence with repeated health outcome measures[71].

Finally, our study participants are mostly representative of the European population since the data was collected from six different European countries. Therefore, caution must be taken when extrapolating our findings to different populations. As more exposome datasets become available in the future, we will be able to further validate the scores obtained in this study and generalize our findings beyond the populations here analyzed. Beyond the validation of our scores, the associations extracted in this study, which are yet rarely studied in exposome research, could be validated more independently without such a high-dimensional dataset. Some of those factors are easily collectable through questionnaires (e.g., noise disturbance, maternal stress for mental health). Social and perceived environmental factors are yet poorly covered in exposome research[72] and this study provides more evidence of their importance.

The combined use of complex machine learning techniques and explainable artificial intelligence methods remains uncommon in the environmental epidemiology field, despite its excellent fit with the exposome paradigm, which aims to capture complex associations within mixtures of environmental factors. Our study revealed results mostly consistent with previous studies, while at the same time, exposing individual level relationships with nonlinearities. Furthermore, we believe that the development of bigger databases and federated analysis tools such as dataShield[73] can unlock the true potential of these approaches to more accurately capture exposome-health associations.

## Conclusion

In this large exposomic study, environmental clinical risk scores were computed using linear (LASSO) and nonlinear methods (XGBoost, Random Forest). No significant differences in predictive performances among these methods were found across the computed risks. From the nonlinear risk scores, we extracted exposome-health relationships from Shapley values, which is, to our knowledge, a novelty in the field, allowing us to derive feature importance at a local and a global level and uncover interactions. The most important predictors included maternal stress, child BMI and noise exposure for mental health; biomarkers such as IL1B and APOA1 for cardiometabolic health; and child BMI and sialic acid (Neu5Ac) for respiratory health.

Besides their usefulness for epidemiological research, our risk scores showed great potential to capture holistic individual-level nonhereditary risk associations that can inform practitioners about actionable factors of high-risk children. As in the post-genetic era, personalized prevention medicine will focus more and more on modifiable factors, we believe that such integrative approaches will be instrumental in shaping future healthcare paradigms.

## Data availability

The raw data supporting the current study, including the proteomics and metabolomics data, are not publicly available due to ethical considerations and legislative restrictions. These data include sensitive information that requires careful handling to protect participant confidentiality and comply with data protection regulations. The access to the raw data is available from the corresponding author on request subject to ethical and legislative review. The "HELIX Data External Data Request Procedures" are available with the data inventory at this website: http://www.projecthelix.eu/data-inventory. The document describes who can apply to the data and how, the timings for approval, and the conditions for data access and publication. Additionally, the data were made available for the exposome data challenge without the feature annotations to fullfill the privacy requirements of the birth cohorts: The datasets are available in the github repository. The source data supporting the study figures can be accessed here[74]. All other data are available, if applicable, from the corresponding author on reasonable request.

## Code availability

Python code is publicly available on GitHub[75].

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

## Acknowledgements

The authors would like to thank all the participating children, parents, practitioners, and researchers in the six countries who took part in this study. The Norwegian Mother, Father and Child Cohort Study is supported by the Norwegian Ministry of Health and Care Services and the Ministry of Education and Research. We also acknowledge the support of the Spanish Ministry of Science and Innovation to the EMBL partnership, the Centro de Excelencia Severo Ochoa, and the CERCA Programme / Generalitat de Catalunya. The CRG/UPF Proteomics Unit is part of the Spanish Infrastructure for Omics Technologies (ICTS OmicsTech) and it is supported by "Secretaria d'Universitats i Recerca del Departament d'Economia i Coneixement de la Generalitat de Catalunya" (2021SGR01225 and 2021SGR01563). This project was funded by the H2020-EU.3.1.2.—Preventing Disease Programme under grant agreement no 874583 (ATHLETE project). JB Guimbaud was supported by a CIFRE PhD fellowship (#2020/1297). Léa Maitre is funded by a Juan de la Cierva-Incorporación fellowship (IJC2018-035394-I) awarded by the Spanish Ministerio de Economía, Industria y Competitividad. ISGlobal and the Exposome hub acknowledges support from the Spanish Ministry of Science and Innovation through the "Centro de Excelencia Severo Ochoa 2019–2023" Program (CEX2018-000806-S), and support from the Generalitat de Catalunya through the CERCA Program. Jose Urquiza is supported by Catalan program PERIS (Ref.: SLT017/20/000119), granted by Departament de Salut de la Generalitat de Catalunya (Spain). Oliver Robinson was funded by the UK

**Article**

Research and Innovation Future Leaders Fellowship (MR/S03532X/1). Silvia Alemany holds a Miquel Servet-I contract (CP22/00026) awarded by the Instituto de Salud Carlos III co-funded by the European Union Found: Fondo Social Europeo Plus, FSE + . Mónica López-Vicente is funded by a Juan de la Cierva-Incorporación fellowship (project IJC2020-045355-I, funded by MCIN/AEI/10.13039/501100011033 and for the European Union NextGenerationEU/PRTR). The data of the cohorts (BiB, EDEN, INMA, KANC, MoBa and RHEA) provided to this research leading to these results has received funding from the European Community's Seventh Framework Programme (FP7/2007-2013) under grant agreement no 308333—the HELIX project. Fig. 2 was created with BioRender.com.

## Author contributions

J.B. Guimbaud: Conceptualization, Methodology, Software, Formal analysis, Data Curation, Writing—original draft, Visualization; A.P. Siskos: Methodology, Resources, Writing—review and editing; A.K. Sakhi: Investigation, Writing—review and rditing; B. Heude: Data Curation; E. Sabidó: Methodology, Writing—review and editing; E. Borràs: Methodology, Formal Analysis, Writing—review and rditing; H. Keun: Writing—Review and Editing; J. Wright: Investigation, Writing—review and editing, Funding Acquisition, Project Administration; J. Julvez: Writing—review and editing; J. Urquiza: Writing—review and editing, Data Curation; K.B. Gützkow: Investigation, Writing—review and editing; L. Chatzi: Writing—Review and Editing; M. Casas: Writing—review and rditing, Data Curation; M. Bustamante: Methodology, Writing—review and editing; M. Nieuwenhuijsen: Writing—Review and Editing; M. Vrijheid: Writing—Review and Editing, Project Administration; M. López-Vicente: Methodology, Data Curation, Writing—Review and Editing; M.C. Pascual: Methodology, Software, Resources, Data curation, Writing—review and editing; N. Stratakis: Writing—review and editing; O. Robinson: Investigation, Writing—Review and Editing; R. Grazuleviciene: Writing—review and editing; R. Slama: Writing—review and editing; S. Alemany: Conceptualization, Methodology, Formal Analysis, Writing—review and editing; X. Basagaña: Writing—review and editing; M. Plantevit: Conceptualization, Methodology, Writing—review and editing; R. Cazabet: Conceptualization, Methodology, Writing—review and editing; L. Maitre: Conceptualization, Methodology, Writing—review and editing, Supervision.

## Competing interests

The authors declare the following competing interests: JB Guimbaud has received PhD fellowship from an industry partnership CIFRE PhD fellowship (#2020/1297, CIFRE are fully financed by the French Ministry of Higher Education, Research and Innovation) and contracted by the private company Meersens. Meersens did not participate in any way in this study. All other authors declare no competing interests.

## Additional information

Jean-Baptiste Guimbaud [1,2,3,4], Alexandros P. Siskos [5], Amrit Kaur Sakhi[6], Barbara Heude[7], Eduard Sabidó[3,8], Eva Borràs[3,8], Hector Keun[5], John Wright[9,10], Jordi Julvez[1,11,12], Jose Urquiza [1,3,11], Kristine Bjerve Gützkow[6], Leda Chatzi[13], Maribel Casas[1,3,11], Mariona Bustamante [1,3,11], Mark Nieuwenhuijsen[1], Martine Vrijheid[1,3,11], Mónica López-Vicente [1,3,11], Montserrat de Castro Pascual [1,3,11], Nikos Stratakis[13], Oliver Robinson[14,15], Regina Grazuleviciene[16], Remy Slama[17], Silvia Alemany [18,19,20], Xavier Basagaña [1,3,11], Marc Plantevit[21], Rémy Cazabet [2] & Léa Maitre[1,3,11] ✉

[1]ISGlobal, Barcelona, Spain. [2]Univ Lyon, UCBL, CNRS, INSA Lyon, LIRIS, UMR5205, F-69622 Villeurbanne, France. [3]Universitat Pompeu Fabra (UPF), Barcelona, Spain. [4]Meersens, Lyon, France. [5]Imperial College London, Cancer Metabolism & Systems Toxicology Group, Division of Cancer, Department of Surgery & Cancer, London, UK. [6]Norwegian Institute of Public Health, Oslo, Norway. [7]Université Paris Cité, Inserm, INRAE, Centre for Research in Epidemiology and StatisticS (CRESS), Paris, France. [8]Centre de Regulació Genòmica, Barcelona Institute of Science and Technology (BIST), Barcelona, Spain. [9]Bradford Institute for Health Research, Bradford, UK. [10]Bradford Teaching Hospitals NHS Foundation Trust, Bradford, UK. [11]CIBER Epidemiología Y Salud Pública (CIBERESP), Madrid, Spain. [12]Institut d'Investigació Sanitària Pere Virgili, Hospital Universitari Sant Joan de Reus, Reus, Spain. [13]Department of Preventive Medicine, University of Southern Los Angeles, Los Angeles, CA, USA. [14]Medical Research Council Centre for Environment and Health, School of Public Health, Imperial College London, London, UK. [15]Mohn Centre for Children's Health and Well-being, School of Public Health, Imperial College London, London, UK. [16]Department of Environmental Sciences, Vytautas Magnus University, Kaunas, Lithuania. [17]Team of Environmental Epidemiology, IAB, Institute for Advanced Biosciences, Inserm, CNRS, CHU-Grenoble-Alpes, University Grenoble-Alpes, Grenoble, France. [18]Psychiatric Genetics Unit, Group of Psychiatry Mental Health and Addiction, Vall d'Hebron Research Institute (VHIR), Universitat Autònoma de Barcelona, Barcelona, Spain. [19]Department of Mental Health, Hospital Universitari Vall d'Hebron, Barcelona, Spain. [20]Biomedical Network Research Centre on Mental Health (CIBERSAM), Instituto de Salud Carlos III, Madrid, Spain. [21]EPITA Research Laboratory (LRE), Kremlin-Bicêtre, France. ✉e-mail: lea.maitre@isglobal.org

