## [Peer Review File · Communications Medicine]

Reviewers' comments:

Reviewer #1 (Remarks to the Author):

In the present paper, Guimbaud et al. analyze the data from 622 mother-child pairs of the HELIX European birth cohorts project. The study aimed to evaluate the effect of more than 300 environmental exposures (including chemical, urban and lifestyle exposures), 100 child peripheral markers (including plasma proteome, serum and urine metabolome) and 18 mother-child clinical markers to compute cross-sectional environmental-clinical risk scores (ECRS) for three health outcomes (child behavioral difficulties, metabolic syndrome, lung function).

The study is innovative and seeks to apply Machine Learning technologies to study the effect of the exposome in modulating health outcomes. In particular, the authors applied LASSO and two non-linear methods (Random forests and XGBoost). I have however some concerns that in my opinion, need to be resolved before the publication of this paper.

First, If I understand correctly the nature of this study is cross-sectional as the time dimension of risk is not accounted for. I would suggest mentioning this limitation in the discussion section.

Abstract:

-the repeated emphasis on the use of “non-linear” methods is not appropriate given that the majority of machine learning methods are non-linear.

- “derive feature importances and interactions”, I think “importance” is uncountable.

Background:

-“ Therefore, there is a need for more advanced machine learning approaches that focus on complex mixtures of exposures and, more specifically, can leverage complex interactions between environmental exposures’ data to generate aggregated measures and capture the main domains of the exposome comprehensively, test its relationship with health measures and facilitate the integration of exposome measures in studies of human development”. There is a need for a suitable conceptual framework for the design and analysis of suitable studies accounting for the biological complexity of the problem. Machine learning is not solving the problem by itself if the data are not collected according to a designed aim.

- “we propose to use advanced machine learning predictive models”, I would suggest removing “advanced”, as these methods are advanced in their nature.

Method:

“Omic data”. I am afraid that Omic is not a suitable term in this context since it involves also exposomics and 122 measures might not be enough to be an “omic” assessment.

Statistical analysis, data preparation “...discarding individuals with more than 50% and exposures with more than 60% missing data to preserve imputation performances”. Please change in “Discarding data from records”. Please do not write “discarding individuals”.

Moreover, how many records were finally discarded?

“Missing values were imputed once using MissForest...”, a full analysis of the robustness of the results with respect to the imputation procedure should be performed. Alternative imputation procedures should be possibly considered.

“Each model considered all exposures and covariates simultaneously in a single multivariate regression. Hyperparameter optimization of the machine learning methods was done using a grid search procedure with a 10-fold cross validation”. Additional details about the optimization procedures should be provided. Non-linear problems are leading to multiple minima in the error surface and problems of model identifiability which should be also discussed.

$Y=f_0(X) +g_0(Z)+R, E[R|Z,X]=0$, There is no information about the distribution of the error term in the model. There is the underlying assumption of Gaussian noise but in which way this was assessed?

Results

“LASSO explained around 12 (standard deviation: 7), 53 (5) and 2% (3) of P-factor, MetS and lung function respectively, whereas RF explained 10 (5), 43 (5) and 2% (2) and XGBoost 12 (5), 51 (5) and 3% (3) variance. Across the three outcomes, the differences between LASSO and XGBoost were not statistically significant (10 fold cross validated two sided paired student t-tests p-values: 0.942, 0.309 and 0.199 for P-factor, MetS and lung function)”. The large part of the variance explained for the Mets score could be related to predictors that are strictly associated with the outcome variable. A clear comparative analysis should be performed to understand the contribution of the single variables to the score looking at the gain of the non-linear modeling procedure with respect to linear modeling approaches.

“The combined use of complex machine learning and explainable AI methods is still quite rare in the field of environmental epidemiology despite its perfect fit with the exposome paradigm, where one seeks to capture complex associations in mixtures of environmental factors. Even if no significant differences in predictive performances with the reference linear method were observed in our study, we believe that the development of bigger databases and federated analysis tools such as dataShield 71 would unlock its potential to capture more accurately the exposome-health associations”. The comparative use of different modeling strategies could be indeed informative. However, the design issue and the role and management of prior hypotheses on causal relations are expected to improve this research much more than the increase in the size of datasets which could be a delicate problem according to the issues of differential random noise and bias.

“Exposome-health relationships were extracted from non-linear risk scores with SHAP, including feature importance at a local and a global level and interactions. Results were mostly consistent with the literature”. SHAP is an added value in this work. The consistency with literature is a good point but the Author should stress the novelty detection and the ways to validate their findings.

“those risk scores have great potential in clinical settings, with the ability to capture holistic individual-level nonhereditary risk associations that can inform practitioners about actionable factors of high-risk patients and provide guidance for prevention and treatment”. The clinical/preventive application of the scores is a theme that is not faced in this paper. I suggest that the Authors moderate their claims.

Reviewer #2 (Remarks to the Author):

This manuscript presents an analysis of environmental determinants of three broad child health outcomes using a variety of environmental exposures and biomarkers. There are many interesting,

commendable, and clever aspects of the analysis that shows the thoroughness of the authors. However, the premise of the study is nebulous as I am not sure how to apply the results of this study. The study reads like a prediction/risk stratification study at times and a causal-oriented study at other times, which is an issue for me as I do not think it can be both. Please find specific comments below:

Major Comments

- Lines 128-130 – “the term prediction refers to the inference of diagnostical risk scores that include cross-sectional factors and do not forecast a future outcome.” – I am struggling with the implications of this sentence and thus, the design of the study, because I am not sure what is the purpose of this study. In all cases, it would seem much easier to conduct clinical tests (e.g. psychometric tests, lipid panels, glucose tolerance tests, pulmonary function tests) than to get hundreds of exposure/biomarker data and infer, with substantial error and uncertainty, the risk for such diseases. The clinical tests, in turn, can also be far more specific than a composite score as are the ones for mental health and MetS. So what value is this study providing?
- Lines 301-306 - Were the covariates treated differently than other predictors? It does not appear that they were used to calculate the ECRS. If they were truly important, should they not be included in the score to make it more accurate?
- In cases of strongly correlated variables – what determined which variables would remain and which variables would be removed? How did this process affect the results and their interpretation (e.g. Shapley values)? This seems to be quite a key point that deserves greater exploration.
- Similar to the point above – it was stated that a pre-selection of features was conducted based on prior knowledge, which begs the question – why not use this as an opportunity for discovery? It seems like the analysis is based on the premise of non-targeted and agnostic large scale discovery and prediction, so why not use all the data at your disposal?
- 10-fold cross-validation is useful to mitigate overfitting, it does not replace the need for external validation (at least a dedicated hold-out sample). Thus, it seems unclear whether the current estimates are generalizable to other populations, which seem to be a limitation. I understand that this was acknowledged in the limitation section, I just wanted to note that it nevertheless is a major weakness in light of the observation that cohort explained most of the variance.
- It is unclear whether data imputation was done before or after the 10-fold CV split. Either way, it seemed like the analysis was done using a single imputation. How intensive were the computation time for these models? If it is not unreasonable, then would it be possible to run multiple imputations (e.g. 5-10 sets of imputations) to properly assess the stability of the results and account for variance in imputation when calculating Shapley confidence intervals. On a related note – the method to calculate Shapley value confidence intervals is done via 10-fold CV, which appear to be similar to bootstrapping confidence intervals. While I think this is perfectly fine approach, I do think there needs to be well over 10X repetitions, analogous to the hundreds/thousands of bootstraps.
- Throughout the results and discussion, the text reads as if the study were examining exposure-outcome relationships in a way that relates to causal understanding (meaning, not predictive). One such example is the paragraph on lines 518-524, where the entire paragraph reads as if the authors were examining direct associations. While I understand that Shapley values can help us understand importance to the prediction/classification score, I am not convinced that they should be interpreted the same as causally-oriented research that explicitly tries to isolate the association of two variables while accounting for potential biases and confounders. This is particularly notable as the discussion and conclusion focused on these specific relationships as opposed to the composite ECRS. If my understanding is incorrect, I am happy to hear a response and reconsider.
- Can the authors please explain how to interpret the numbers for interactions? (lines 552-578).

Perhaps this is my own ignorance, but it is unclear what value or threshold constitutes as important when examining absolute SHAP values. Related to this, can the authors discuss the importance for these interactions if LASSO performed equally, if not better?

- It is unclear to me how I should interpret the stratification analysis based on PRS when they explain so little of the disease (5% and 1%, per methods section). So for example, for children at 90% of genetic liability, are they truly as “high” risk? (language taken from line 600). Furthermore, I am not sure I understand how this analysis answers the question of whether they provide complementary information, as opposed to conflicting information. Would it be better to use something like a C-statistic/AUC to address this question? (e.g. comparing PRS only, ECRS only, and both). I acknowledge that this may be a reflection of ignorance on my behalf, but some clarification may be necessary for myself and potential readers.

Minor Comments

- Re: mental health - while a singular composite score (i.e. “p”) is attractive in its simplicity to model, one limitation of this value is that it is a latent representation that reflects common features shared by many psychiatric disorders and is not specific to any. This is not critical, though it is a limitation.
- To aid readers, can the authors distinguish between prenatal vs. postnatal factors in the results? For both the text as well as figures? Right now, it is difficult to discern which factors were prenatal vs. postnatal.
- Lines 644-645 – would the consistency not simply reflect the pre-selection of features based on prior knowledge? Would any possible result have been largely inconsistent?
- I am not sure that being able to explain 12% and 3% of variance using 500+ predictors is important for clinical settings (abstract and conclusions section).

Reviewer #3 (Remarks to the Author):

This manuscript investigates the value of environmental-clinical risk scores (ECRS) for predicting three types of health outcomes: mental health, cardiometabolic syndrome, and respiratory health. The study uses data from the HELIX project that includes data from six different European longitudinal birth cohorts. The results suggest that ECRS captures 12%, 51%, and 3% of the variance in mental, cardiometabolic, and respiratory health respectively. Predictive methods employed were LASSO, XGBoost, and Random Forests.

While the study is interesting, there are several major concerns about the manuscript in its current form. The modeling aspects are poorly written and organized. The manuscript fails to make effective use of figures and tables to clarify its methodology. Several of these issues as well as other issues are pointed out below.

1. A detailed data table is needed to identify the multiple cohorts and indicate the variable types available for each cohort, which type of mental health outcome they are being used to predict, and what percent of the data is incomplete.
2. It takes a lot of effort to understand that effectively there are three different prediction problems with mental health, cardiometabolic syndrome, and respiratory health outcomes as outputs and environmental factors as inputs. Fig. 2 which describes workflow fails to clearly point out what the model inputs and outputs are.
3. On the one hand, the paper treats ECRS like a simple scalar measure similar to PRS that can be used to assess environmental risk given a fixed health outcome. On the other hand, the environmental parameters seem to be inputs to the models. If multivariate environmental inputs are

used, where does the ECRS come in and how is it computed? All classification models presented here take in multiple inputs and try to predict a single health outcome as an output. Then where does a single ECRS variable corresponding to each outcome come in?

4. Equally unclear are the roles of PRS, omics, and clinical variables. Are these model inputs? How do they relate to the ECRS?

5. LASSO is repeatedly referred to as a linear technique. LASSO could use a linear or generalized linear model. But due to the L1 constraint, LASSO estimation itself is not a linear approach. This needs to be clarified.

6. Since the HELIX study includes data from six different cohorts, it is surprising that the cohort index was not treated as a covariate to compensate for cohort-specific bias. Not only do the cohort indices need to be treated as covariates in the model, the authors should examine performance differences and interpretation differences that may exist across cohorts/data sources.

Point-By-Point Response to reviewers

Reviewer #1:

Remarks to the Author:

In the present paper, Guimbaud et al. analyze the data from 1622 mother-child pairs of the HELIX European birth cohorts project. The study aimed to evaluate the effect of more than 300 environmental exposures (including chemical, urban and lifestyle exposures), 100 child peripheral markers (including plasma proteome, serum and urine metabolome) and 18 mother-child clinical markers to compute cross-sectional environmental-clinical risk scores (ECSR) for three health outcomes (child behavioral difficulties, metabolic syndrome, lung function).

The study is innovative and seeks to apply Machine Learning technologies to study the effect of the exposome in modulating health outcomes. In particular, the authors applied LASSO and two non-linear methods (Random forests and XGBoost). I have however some concerns that in my opinion, need to be resolved before the publication of this paper.

First, If I understand correctly the nature of this study is cross-sectional as the time dimension of risk is not accounted for. I would suggest mentioning this limitation in the discussion section.

We appreciate the reviewer's positive assessment of our study. While this is true that our ECSR are not meant to forecast future outcomes, the study is not entirely cross-sectional as, in addition to childhood cross-sectional factors, we included numerous variables (n=62) that were measured during pregnancy (full description in Supplementary Tables 3.2 and 3.3). To avoid unnecessary confusion, we removed the cross-sectional qualification of our study in the abstract (line 61) and reformulated the presentation of our approach in the Introduction (line 146-147) as well as in the Methods-Data section (line 193-194). As our ECSR do not capture any temporal relationships, we added this as a limitation in the discussion section (lines 895-905) as such:

"Another limitation is that our study does not account for the risk evolution over time since we used pregnancy and cross-sectional epidemiological data to assess childhood exposures at a single point in time, without considering their impact throughout an individual's lifetime. This limitation will be addressed in ATHLETE, the HELIX follow-up project in which the same children were regularly monitored into adolescence with repeated health outcome measures²".

Abstract:

- The repeated emphasis on the use of "non-linear" methods is not appropriate given that the majority of machine learning methods are non-linear.

We took the comment into consideration and reduced the emphasis on the use of "non-linear machine learning" methods throughout the manuscript (lines 53, 438). We did not completely removed it as one of our primary objectives was to use an approach to compute nonlinear risk scores and derive nonlinear relationships, which is rare in the field.

- "derive feature importances and interactions", I think "importance" is uncountable.

We apologize for the mistake and have deleted all instances of "s" for *feature importance* throughout the manuscript.

Background:

-“ Therefore, there is a need for more advanced machine learning approaches that focus on complex mixtures of exposures and, more specifically, can leverage complex interactions between environmental exposures’ data to generate aggregated measures and capture the main domains of the exposome comprehensively, test its relationship with health measures and facilitate the integration of exposome measures in studies of human development”. There is a need for a suitable conceptual framework for the design and analysis of suitable studies accounting for the biological complexity of the problem. Machine learning is not solving the problem by itself if the data are not collected according to a designed aim.

We agree that machine learning techniques are complementary tools, not standalone solutions, that can help to analyze complex biological systems. By highlighting the necessity for advanced machine learning approaches, we wanted to point out that traditional statistical methods may not effectively capture the intricate relationships within complex mixtures of environmental exposures. Our statement was not meant to diminish the importance of well-designed studies, informed by a solid conceptual understanding of the biological complexity of the problem. We added a sentence in the main text to clarify this point (lines 109-113) such as:

Therefore, there is a need to collect a wide range of environmental data coupled with clinical biomarkers to comprehensively capture the main domains of the exposome before the apparition of clinical symptoms and apply more advanced modeling approaches suitable for complex mixtures of exposures that can leverage their interactions to facilitate its integration in observational and clinical studies.

- “we propose to use advanced machine learning predictive models”, I would suggest removing “advanced”, as these methods are advanced in their nature.

We rewrote this specific paragraph to expose the study rationale more clearly. However, following the reviewer suggestion, we replaced the term “advanced” with “recent” in the new paragraph (line 152). For the record, we used the term “machine learning” to refer to any method using automated learning, whether complex or simple, including linear methods. In that regard, we used the term “advanced” to highlight that the methods that we used can capture more complex nonlinear relationships.

Method:

“Omic data”. I am afraid that Omic is not a suitable term in this context since it involves also expos-omics and 122 measures might not be enough to be an “omic” assessment.

We appreciate the reviewer’s remark and decided to replace the term omic with (pre)clinical biomarkers.

Statistical analysis, data preparation “...discarding individuals with more than 50% and exposures with more than 60% missing data to preserve imputation performances”. Please change in “Discarding data from records”. Please do not write “discarding individuals”. Moreover, how many records were finally discarded?

We replaced “individuals” with “records of individuals” (line 351), and also included the number of discarded records (102 records and 3 variables).

“Missing values were imputed once using MissForest...”, a full analysis of the robustness of the results with respect to the imputation procedure should be performed. Alternative imputation procedures should be possibly considered.

As suggested, we have further evaluated the results with other imputation methods. We compared the performances of three single imputation methods (i.e., MissForest, KNN, and mean imputation) by introducing additional missing values in our data. We added the performances obtained with each method in the Methods section (lines 360-364) as such:

“We compared performance of this method with two classical single imputation methods, namely KNN and mean imputation on manually generated missing values. The number of missing values to add on top of the originals was settled to be proportional to 15% of the total sample size for each variable (i.e., 243). MissForest considerably outperformed KNN and mean imputation with a mean squared error of 0.56, 1.00 and 1.25 respectively.”

“Each model considered all exposures and covariates simultaneously in a single multivariate regression. Hyperparameter optimization of the machine learning methods was done using a grid search procedure with a 10-fold cross validation”. Additional details about the optimization procedures should be provided. Non-linear problems are leading to multiple minima in the error surface and problems of model identifiability which should be also discussed.

We thank the reviewer for highlighting this point. Following a comment of reviewer 3, we reformulated the Methods – Modeling section to clarify the study workflow. In this new paragraph, we explained more clearly the hyperparameters’ (HP) tuning procedure inside the cross-validation (CV) loop and the choice of the CV procedure (lines 381-392). Full details about the HP optimization are displayed in **Supplementary Tables 3.6-3.7**, with selected HP, limits of the tested domain and random seed used, which ensures full reproducibility of our results.

Concerning the reviewer’s comment on model identifiability, we added a mention to the problem in the manuscript and stated that both Random Forest and XGBoost hyperparameters were optimized through cross-validation, which aids in avoiding overfitting and, indirectly, in addressing the problem (lines 766-774) as such:

“A known limitation of most nonlinear machine learning approach is the model identifiability which can lead to complications such as overfitting or ambiguity in interpreting the model’s parameters^{3,4}. However, we believe our approach has several strengths that mitigate these concerns. First, our modeling strategy, backed by rigorous cross-validation, limits overfitting and prioritizes generalizability across different data subsets. Second, both nonlinear methods used in this study aggregates results from multiple models, whether from boosting or bagging, which enhance the stability of results. Finally, it is worth remembering that identifiable models such as linear regression or LASSO are also limited. They cannot capture nonlinear relationships and their interpretation are correct only if all relationships are linear.”

$Y=f_0(X) +g_0(Z)+R, E[R|Z,X]=0$, There is no information about the distribution of the error term in the model. There is the underlying assumption of Gaussian noise but in which way this was assessed?

With $E[R|Z,X]=0$, we assumed the expected values of the residuals R to be 0, meaning that there is no systematic bias in the models. We added information on the residuals in the main text (lines 541-544) and in **Supplementary Table S3.8**, namely about their mean and the nature of their distributions across all folds of the cross-validation procedure. However, since we used XGBoost for our final ECRS, the normality assumption is not required.

Results

“LASSO explained around 12 (standard deviation: 7), 53 (5) and 2% (3) of P-factor, MetS and lung function respectively, whereas RF explained 10 (5), 43 (5) and 2% (2) and XGBoost 12 (5), 51 (5) and 3% (3) variance. Across the three outcomes, the differences between LASSO and XGBoost were not statistically significant (10 fold cross validated two sided paired student t-tests p-values: 0.942, 0.309 and 0.199 for P-factor, MetS and lung function)”. The large part of the variance explained for the Mets score could be related to predictors that are strictly associated with the outcome variable. A clear comparative analysis should be performed to understand the contribution of the single variables to the score looking at the gain of the non-linear modeling procedure with respect to linear modeling approaches.

To highlight what part of the MetS score is attributable to the Metabolites/Proteins markers (i.e., the predictors that are strictly associated with cardiometabolic health) and the others variables (environmental exposures, clinical factors and covariates), we computed shapley values for those separate groups of exposures (**Supplementary Figure S3.3**) and discuss it in the Results section (lines 554-556) and in the discussion (lines 889-891). We also added a color code to **Figure 5** to visually separates factors that are Metabolites/Proteins from clinical factors, exposures and covariates. While it is clear that metabolites and proteins explains most of the cardiometabolic ECRS, exposure variables have still a significant impact (mean SHAP 0.50 and 0.14 for metabolites/proteins and exposures respectively), notably lifestyle and build environment exposures.

Concerning the second point raised by the reviewer, in addition to predictive performances, we slightly extended the scope of our comparative analysis by extracting Shapley values for Lasso. Results are available in **Supplementary figures S3.4-3.5** and described in Results such as:

“Compared with the linear associations obtained with Lasso (**Supplementary Figure S3.5**), directions of associations were consistent with XGBoost, with some exceptions (e.g., leptin for cardiometabolic health and bus line accessibility for respiratory health). Overall, predictions obtained with XGBoost were more conservative for extreme values of the predictors (e.g., maternal stress for mental health or child zBMI for respiratory health, etc.).”

“The combined use of complex machine learning and explainable AI methods is still quite rare in the field of environmental epidemiology despite its perfect fit with the exposome paradigm, where one seeks to capture complex associations in mixtures of environmental factors. Even if no significant differences in predictive performances with the reference linear method were observed in our study, we believe that the development of bigger databases and federated analysis tools such as dataShield would unlock its potential to capture more accurately the exposome-health associations”. The comparative use of different modeling strategies could

be indeed informative. However, the design issue and the role and management of prior hypotheses on causal relations are expected to improve this research much more than the increase in the size of datasets which could be a delicate problem according to the issues of differential random noise and bias.

Our study's primary aim is to explore potential exposome-health relationships using a novel approach in the field. We do not intend to study nor confirm the causal links between these relationships and therefore, did not include any prior hypothesis on causal pathways. This was made more explicit in the Discussion (lines 783-787 and 880-882). We shortened the part of the Discussion about causal pathways in the main text (we deleted paragraphs lines 829-851), and we more explicitly expose unexpected and poorly literature covered relationships (lines 789-827) to inform future research through hypothesis generation. Additionally, in the Discussion section, we acknowledged the fact that our data only included exposures measured during pregnancy and childhood, and that the temporality of exposure is not considered in our study (lines 895-905). We also mentioned that temporality would be addressed in future work (ATHLETE) using data from the follow-up cohort into adolescence (12-18 years old).

About the other point that the reviewer is raising, it is true that besides giving more statistical power, having more data can also increase the chance of capturing spurious associations if the data are noisy and especially if no domain knowledge was used to tailor the captured relationships (such as in our exploratory study). In the general case however more data, especially on unseen populations, gives additional information for the model to learn the true distribution and to better generalize. Additionally, simple methods, such as linear regression (LASSO, etc.), are known to perform better than more complex models on small datasets since they are less likely to overfit on those data (c.f. Hastie and al, 2009, chapter 2.9 p.37). Although it is possible that the relationships to be captured were mostly linear, it is, in our opinion, important to note that the size of the dataset used in this study is likely to explain the comparable performance between LASSO and methods such as XGBoost. We added a more explicit mention of that concern in the main text (lines 776-787). We also wanted to emphasize the importance in the future of easier access to data from multiple populations (with methods such as federated learning) to unlock the potential of complex modeling approaches like the one we used in this study.

“Exposome-health relationships were extracted from non-linear risk scores with SHAP, including feature importance at a local and a global level and interactions. Results were mostly consistent with the literature”. SHAP is an added value in this work. The consistency with literature is a good point but the Author should stress the novelty detection and the ways to validate their findings.

We thank the reviewer for this positive feedback. We emphasized more on the novelty aspect of SHAP in the conclusion (lines 930-933), and, while we shortened the discussion on previously observed relationships in the literature (lines 829-851) we also included short discussions on observed relationships with strong or poor literature coverage to this day (lines 789-827). Additionally, while previous studies have documented the significant roles of exposures, such as maternal stress and noise, in shaping child preclinical symptoms; our study provides a more comprehensive exploration of the myriad of biological mechanisms at play by allowing for the simultaneous assessment of their interactions and cumulative effects.

Concerning the validation of our findings, we weren't able to find access to external birth cohort data sufficiently close in terms of variety of factors to those used in this study. Instead, we chose to leverage the six different cohorts available in our data to perform a cohort-based cross-validation (see Methods, lines 478-484; Results, lines 693-701; and Discussion, lines 706-708). Consistent results were achieved across cohorts, yielding promising results for the generalizability of our scores on European population. However, we acknowledge that a validation on an external dataset with non-European origins would be valuable. We now discuss it with more details (lines 906-915) as such:

“Finally, our study participants are mostly representative of the European population, since the data was collected from six different European countries. Therefore, caution must be taken when extrapolating our findings to different populations. As more datasets became available in the future, we will be able to further validate the scores obtained in this study and generalize our findings beyond the populations here analyzed. Beyond the validation of our scores, the associations extracted in this study, that are yet rarely studied in exposome researches, could be validated more independently without such a high dimensional dataset. Some of those factors are easily collectable through questionnaires (e.g., noise disturbance, maternal stress for mental health). Social and perceived environmental factors are yet poorly covered in exposome research ⁵ and this study provides more evidence of their importance.”

“those risk scores have great potential in clinical settings, with the ability to capture holistic individual-level nonhereditary risk associations that can inform practitioners about actionable factors of high-risk patients and provide guidance for prevention and treatment”. The clinical/preventive application of the scores is a theme that is not faced in this paper. I suggest that the Authors moderate their claims.

We removed the mentioned sentence in both the Conclusion and the Abstract and replaced it with:

“Besides their usefulness for epidemiological research, our risk scores showed great potential to capture holistic individual-level nonhereditary risk associations that can inform practitioners about actionable factors of high-risk children. As in the post-genetic era, personalized prevention medicine will focus more and more on modifiable factors, we believe that such integrative approaches will be instrumental in shaping future healthcare paradigms.”

Reviewer #2:

Remarks to the Author:

This manuscript presents an analysis of environmental determinants of three broad child health outcomes using a variety of environmental exposures and biomarkers. There are many interesting, commendable, and clever aspects of the analysis that shows the thoroughness of the authors. However, the premise of the study is nebulous as I am not sure how to apply the results of this study. The study reads like a prediction/risk stratification study at times and a causal-oriented study at other times, which is an issue for me as I do not think it can be both. Please find specific comments below:

We thank the reviewer for the positive assessment of our study as well as the relevant comment. We significantly reduced the discussion of causal pathways in the main text (see

answer to specific comments) and emphasized the exploratory purpose of our study. The discussion on literature-based marginal relationships aimed to demonstrate that our approach, which captured more complex relationships than those typically used in the field, yielded results that were largely in line with the existing literature (comparing comparable elements such as directions of associations). It also aimed at highlighting which relationships are well known and which would benefit from further research. In that regard we reduced the discussion on the interpretation of relationships extracted in our study and focus more on their coverage in the literature (lines 789-851). While relationships identified in our study are not necessarily causal, they still provide a valuable foundation for future confirmatory causal analyses. We therefore believe it is important to discuss and highlight these findings. Following the reviewer's comment, we rewrote the study rationale to more explicitly state our aims (lines 148-162).

Major Comments

•Lines 128-130 – “the term prediction refers to the inference of diagnostic risk scores that include cross-sectional factors and do not forecast a future outcome.” – I am struggling with the implications of this sentence and thus, the design of the study, because I am not sure what is the purpose of this study. In all cases, it would seem much easier to conduct clinical tests (e.g. psychometric tests, lipid panels, glucose tolerance tests, pulmonary function tests) than to get hundreds of exposure/biomarker data and infer, with substantial error and uncertainty, the risk for such diseases. The clinical tests, in turn, can also be far more specific than a composite score as are the ones for mental health and MetS. So what value is this study providing?

We thank the reviewer for this question which allowed us to better clarify our objectives and the value of our scores. While clinical tests have undeniable value in disease diagnosis and treatment, they also have limitations (e.g., expensive to perform, require access to a medical facility, struggle to address multifactorial causes) that could be circumvented with multifactorial risk score analysis. In particular, composite risk scores can provide a more comprehensive understanding of risk factors than clinical tests, especially for diseases with complex etiologies. Therefore, they can have broader implications for public health policy by identifying key actionable environmental or pre-clinical risk factors. In addition, composite risk scores can be used to help identify at-risk populations and encourage them to undergo informed clinical tests. While we acknowledge the uncertainties associated with our ECRS, we believe that our approach could provide an additional layer of understanding of disease risk and complement traditional diagnostic tools. We extended the presentation and justification of the ECRS value in the introduction (lines 120-130).

Another important point is that current clinical tests fail to identify at-risk children before the onset of chronic diseases, especially mental and cardiovascular diseases, which is key to effective prevention and treatment policies. This is largely attributed to the fact that many of these tests were primarily developed and validated in adults, and to the paucity of longitudinal data on children from the general population. In the case of cardiovascular risk, the role of adipokines is a pertinent example: while the association between elevated inflammatory biomarkers and increased cardiovascular risk is well established in adults, comprehensive studies on the onset and progression of these biomarkers in children is lacking⁶. Therefore, to provide guidance to the pediatric medical community for more effective interventions and prevention policies, there is a need for more prospective studies focused on children, such as

HELIX and ATHLETE. We added this reflection in the manuscript's discussion section (lines 735-745) as such:

“Our study was performed in a context where current clinical tests often fail to identify children at-risk of developing chronic diseases, notably considering mental and cardiovascular diseases, which is a key challenge to the development of effective prevention and treatment policies. This limitation is largely attributed to the fact that many of these tests were developed and validated primarily in adult populations and to the paucity of longitudinal data on children. In the case of cardiovascular risk, the role of adipokines is a pertinent example. While the association between elevated inflammatory biomarkers and increased cardiovascular risk is well established in adults, there is a lack of comprehensive studies on the onset and progression of these biomarkers in children ⁶. Therefore, more prospective studies focused on children are necessary to provide guidance to the pediatric medical community for more effective interventions and prevention policies.”

•Lines 301-306 - Were the covariates treated differently than other predictors? It does not appear that they were used to calculate the ECRS. If they were truly important, should they not be included in the score to make it more accurate?

We appreciate the reviewer bringing this ambiguity to our attention. We used covariates in the scores with the other predictors, which we clarified in the main text (Method – Data – Covariates section, line 278).

• In cases of strongly correlated variables – what determined which variables would remain and which variables would be removed? How did this process affect the results and their interpretation (e.g. Shapley values)? This seems to be quite a key point that deserves greater exploration.

We thank the reviewer for this important remark. One primary goal in this study was to discover non-targeted exposome-health relationships, which, without further explanations, seems incompatible with the multiple layers of data selection we performed (namely an expert-driven pre-selection, that is described in the next reviewer comment, and a data-driven selection, described in the next paragraph). Those layers of selection were kept minimal in order to find a correct balance between agnostic discovery and optimization of both predictive performances and models interpretability.

The description and explanation of selections steps were added in the Methods – Statistical Analysis – 1. Data preparation section (lines 329-348). In cases where variables were strongly correlated, in order to decrease multicollinearities in the dataset, increase model interpretability, and reduce the likelihood of overfitting, we performed the following selection process: 1) if the correlation was between the same variables averaged over different time frames (e.g., day, week, year), we kept the one with the longest period; 2) if the correlation was between the same variables computed at home and at school, or other places, we kept the variable computed at home. These rules were designed to retain features that are more informative and universally applicable (e.g., home and school exposures). For the other cases, the default rule was to keep the first variable in the order in which they appeared. We added a description of the data-driven variable selection rules in **Annex 2**. We could have applied other rules (such as selecting the variable most correlated with the outcome), but since they apply only to groups of very strongly correlated variables (Pearson $r > 0.9$) (meaning that the

remaining variable still encodes most of the information), this selection step is likely to have little impact on the model's predictive performance.

•Similar to the point above – it was stated that a pre-selection of features was conducted based on prior knowledge, which begs the question – why not use this as an opportunity for discovery? It seems like the analysis is based on the premise of non-targeted and agnostic large scale discovery and prediction, so why not use all the data at your disposal?

We performed the pre-selection before the analysis. For this, we selected all the usable data available for the HELIX subcohorts and included new variables previously unexplored in the HELIX studies (lines 228-232). Although, we could have included more variables, we chose to prioritize model simplicity, interpretability, or contextual relevance. We selected representatives from groups of related variables. For instance, we only kept the 300m buffers for the number of road intersections per km² and removed the 100m buffers; also, we kept only home-based air pollutants measurements and discarded those from school or other places. As stated in the precedent comment, we added a description in the main text (line 329-349) and the additional details are provided in **Annex 2**.

We acknowledge the potential value of an exploratory analysis using all variables however, our work, while aiming to achieve large-scale, agnostic discovery, was also guided by other important considerations that included the maximization of the predictive capabilities of derived ECRS and their interpretability. This approach aimed to achieve a balance between the exploratory goals and the need for interpretability and performance.

•10-fold cross-validation is useful to mitigate overfitting, it does not replace the need for external validation (at least a dedicated hold-out sample). Thus, it seems unclear whether the current estimates are generalizable to other populations, which seem to be a limitation. I understand that this was acknowledged in the limitation section, I just wanted to note that it nevertheless is a major weakness in light of the observation that cohort explained most of the variance.

We agree with the reviewer; however, at the moment, we were unable to find external data that closely matched those used in our study (which has the particularity of being particularly rich in terms of the diversity of measured exposures and clinical biomarkers in children). Instead, without changing the workflow of our study, we added a cohort based cross-validation procedure, in which we iteratively computed ECRS on five cohorts to predict the sixth (see Methods, lines 478-484; Results, lines 693-701; and Discussion, lines 706-708) using the same hyperparameters than those used for the final ECRS. Results were consistent across cohorts.

Another method would have been to propose final ECRS trained on five cohorts and keep the sixth for validation. However, this suppose to: 1) arbitrary select a cohort on which performing the validation (or considerably complicates the study by proposing 6 ECRS instead for each outcome instead of one), and 2) loose statistical power for the final ECRS as the sample size used for training would be smaller. Hence, we chose the abovementioned solution.

•It is unclear whether data imputation was done before or after the 10-fold CV split. Either way, it seemed like the analysis was done using a single imputation. How intensive were the computation time for these models? If it is not unreasonable, then would it be possible to run multiple imputations (e.g. 5-10 sets of imputations) to properly assess the stability of the results and account for variance in imputation when calculating Shapley confidence intervals. On a related note – the method to calculate Shapley value confidence intervals is done via 10-fold CV, which appears to be similar to bootstrapping confidence intervals. While I think this is a perfectly fine approach, I do think there needs to be well over 10X repetitions, analogous to the hundreds/thousands of bootstraps.

We thank the reviewer for bringing this to our attention. We have now clarified that we performed data imputation before the 10-fold CV procedure which is shown in **Figure 2**. The single imputation procedure was not very costly and only required few hours (6-10). While generating multiple imputed datasets and re-running the whole analysis on each dataset would be feasible, it would require a considerable amount of time and effort given the high cost of hyperparameter tuning for the machine learning models. Additionally, since our study is exploratory in nature and does not aim to assess causal relationships, we believe that using single imputation is sufficient (especially given the relatively low amount of missing data) and easier to understand and replicate. Nevertheless, we performed a sensitivity analysis to compare the imputation performance of MissForest with classical single imputation methods such as KNN and mean imputation. The results of this analysis are reported in the main text (lines 360-364).

Similarly, we chose 10-fold cross-validation to achieve a balance between computational efficiency and our performance's robustness. Using Shapley values estimates as a leave-one-out cross-validation procedure would be very costly and yield little benefit. With 10-fold cross-validation, each fold is large enough to reasonably represent the entire dataset, and thus offers a good trade-off between bias and variance (compared to LOOCV that tends to have higher variance (Hastie and al, 2009, section 7.10.1)).

•Throughout the results and discussion, the text reads as if the study were examining exposure-outcome relationships in a way that relates to causal understanding (meaning, not predictive). One such example is the paragraph on lines 518-524, where the entire paragraph reads as if the authors were examining direct associations. While I understand that Shapley values can help us understand importance to the prediction/classification score, I am not convinced that they should be interpreted the same as causally-oriented research that explicitly tries to isolate the association of two variables while accounting for potential biases and confounders. This is particularly notable as the discussion and conclusion focused on these specific relationships as opposed to the composite ECRS> If my understanding is incorrect, I am happy to hear a response and reconsider.

We appreciate the reviewer's thoughtful comment on the distinction between predictive modeling and causal inference. We agree that the primary focus of our study is on predictive accuracy rather than on causal relationships. Indeed, although Shapley values offer valuable insights into feature importance for prediction, they are not designed to indicate causality. However, the relationships extracted from our ECRS are valuable as a preliminary step in an exploratory context, which needs validation through future confirmatory research. For this reason, we discussed the relationships we obtained from SHAP and compared them with

those observed in the literature. Considering the reviewer comment, we now shortened the discussion concerning these relationships in the main text and made explicit that they are not causal (lines 789-851). Additionally, the limitation section now explicitly states that any relationships extracted from this study require further investigation using a proper causal inference methodology (lines 880-882). Finally, we replaced occurrences of the term “effect” (e.g., strong protective effect), which implies causality by “impact” in the Results and Discussion sections.

•Can the authors please explain how to interpret the numbers for interactions? (lines 552-578). Perhaps this is my own ignorance, but it is unclear what value or threshold constitutes as important when examining absolute SHAP values. Related to this, can the authors discuss the importance for these interactions if LASSO performed equally, if not better?

In our study, the SHAP values for interaction terms aim to measure the degree to which the interaction contributes to the prediction beyond the effect of the individual features. While there is no universal threshold for importance, higher absolute SHAP values typically imply a stronger contribution to the model. To obtain an estimate of their average weights in predictive models, mean absolute amplitudes of pairwise interactions can be compared with mean absolute amplitudes of marginal effects. In this study, interaction amplitudes were small in comparison with marginal effects (7.4 to 8 times smaller), indicating their overall low impact on the predicted ECRS. We made this point more explicit in the manuscript (lines 641-642).

For cases of ECRS where LASSO (without interactions) outperforms the nonlinear model with interactions, it may suggest that exposome-health relationships are linear and that there are no interactions to be captured. However, the difference in prediction is likely to be due to the ability of simpler models to perform better in situations of small training sample size (Hasie et al, 2009, chapter 2). In that case, the relationships to be captured could be nonlinear with interactions among the variables, but we would not have enough data to accurately capture them. In other words, even if LASSO performs well, it may not capture all the complexities within the data, especially nonlinear relationships and interactions, which other algorithms could reveal. Exposing those relationships can reveal interesting relationships within the data that are not evident when considering each feature independently, and would be valuable for generating hypotheses for future research. We added this point in the Discussion section (lines 776-787) as such:

“Our tree-based approach has shown comparable predictive performances to LASSO, which could indicate that the relationships to capture are mostly linear in nature and that there are no interactions to be captured. However, the difference in prediction is likely to be due to the ability of simpler models to perform better in situations of small training sample size⁶². More data would be needed in order to confirm the nature of those relationships and, even if LASSO performs well, it may not capture all the complexities within the data, especially nonlinear relationships and interactions, which other algorithms could reveal. In this study, we explored such relationships using a novel approach in this field. We did not aim to assess nor confirm the causality of extracted relationships, but rather to identify new associations that previous methods would have missed or refine the potential nature of known ones (e.g non linearities, interactions). Thus, the causality of those findings needs to be validated in a causal inference framework”

•It is unclear to me how I should interpret the stratification analysis based on PRS when they explain so little of the disease (5% and 1%, per methods section). So for example, for children at 90% of genetic liability, are they truly as “high” risk? (language taken from line 600). Furthermore, I am not sure I understand how this analysis answers the question of whether they provide complementary information, as opposed to conflicting information. Would it be better to use something like a C-statistic/AUC to address this question? (e.g. comparing PRS only, ECRS only, and both). I acknowledge that this may be a reflection of ignorance on my behalf, but some clarification may be necessary for myself and potential readers.

We thank the reviewer for this pertinent comment. We agree that in this context, PRS are not reliable to identify individuals at risk and hence may not really reflect their true genetic liability. Our idea was that the stronger the correlation between PRS and ECRS is, the higher the likelihood that the information in both the exposome and the genome is redundant. For instance, a perfect correlation would indicate that because PRS and ECRS are identical, neither of the risk scores captures additional information useful for the predictive task. Conversely, no correlation at all would imply that the two scores are independent and capture different aspects of the risks. Additionally, correlation shows that children with high PRS may also be at high risk of other risk scores such as ECRS. In our study, however PRS captured only a small amount of phenotypic variation, which may explain the observed relatively small correlation and complexify the interpretation of our results. Consequently, we chose to remove the PRS analysis from the manuscript (lines 468-477, 679-692, 751-756 and 1254).

We applied the experimental design suggested by the reviewer. However, it faces the same problem regarding the PRS’s poor performance. We performed the analysis and found very minor improvements using both PRS and ECRS (<1% variance explained) compared with ECRS alone. Given the poor predictive performance of PRS, we chose not to include the analysis in the manuscript, as it would complexify the understanding of our methodology with no significant benefits.

Minor Comments

•Re: mental health - while a singular composite score (i.e. “p”) is attractive in its simplicity to model, one limitation of this value is that it is a latent representation that reflects common features shared by many psychiatric disorders and is not specific to any. This is not critical, though it is a limitation.

Phenotypic P-factor is a reflection of the correlation structure across symptom scales and the comorbidity of psychiatric disorders. Therefore, while, as the reviewer is suggesting, by using the P-factor as outcome, we may miss potential associations between specific risk factors and symptoms, research supports that P-factor may capture part of the complexity of psychiatric comorbidity⁹. It is particularly important in children, for whom symptoms are not as differentiated and often shift from one domain to another^{10,11}. In a more general point of view, we advocate that general health risk scores (e.g., P-factor, MetS, etc.) and specific risk scores are both valuable in different settings. General risk scores are less precise than a specific one for a targeted outcome but are easier to use in a preventive setting to identify individuals at risk for whom more specific tests would be needed.

•To aid readers, can the authors distinguish between prenatal vs. postnatal factors in the results? For both the text as well as figures? Right now, it is difficult to discern which factors were prenatal vs. postnatal.

We appreciate the reviewer's feedback. We added * after each pregnancy variable in both **Figure 5** and **Figure 6** to distinguish them explicitly. We have included additional details about each pregnancy variable in the main text.

•Lines 644-645 – would the consistency not simply reflect the pre-selection of features based on prior knowledge? Would any possible result have been largely inconsistent?

We agree with the reviewer that, in principle, a full discovery approach would have been more likely to yield unattended results. However, expert knowledge-based pre-selection only refined selection process among closely related variables, without impacting the diversity of the included variables. The final selection of variables was intentionally broad and enabled the discovery of new relationships.

For the 10 most important factors in each ECRS, most relationships were consistent, in terms of direction of association, with the literature. The inverse distance to the nearest road during pregnancy is unexpectedly associated with higher FEV1, this was however already reported in previous a HELIX study ⁷ and identified to be driven by the RHEA cohort. Other than that, no direction of association was unexpected but some relationships are yet poorly covered. This is described in the Discussion section (lines 789-827).

•I am not sure that being able to explain 12% and 3% of variance using 500+ predictors is important for clinical settings (abstract and conclusions section).

While mental and respiratory ECRS performances are not yet sufficient to be used as a screening tool alone, they still provide value through the identification of captured associations, that, combined with future research would help to disentangle the combined effects of such mixture of exposures. This refined understanding would be beneficial in clinical settings though, for instance, a better education of healthcare practitioners. Moreover, the potential of combined risk scores, such as ECRS to help clinical-decision making has been promoted in various domains of research ^{8,12}, and anticipated to be part of a next generation early intervention infrastructure ¹³. We refined the benefits of environmental risk scores for clinical settings in the Introduction (lines 123-130) such as:

“Unlike genetics, some environmental factors are actionable, which gives ERS a broader potential for shaping public health policies by identifying actionable key factors and facilitate the implementation of preventive and personalize medicine measures. Combined with PRS, these scores could also serve as an initial step in identifying at-risk populations, who can then be directed to more specific clinical diagnostic tests, thereby serving as a complementary tool in the healthcare decision-making process.”

Acknowledging the reviewer's comment, we moderated our claims and refined the final parts of the abstract and the conclusion as such:

"Besides their usefulness for epidemiological research, our risk scores showed great potential to capture holistic individual-level nonhereditary risk associations that can inform practitioners about actionable factors of high-risk children. As in the post-genetic era, personalized prevention medicine will focus more and more on modifiable factors, we believe that such integrative approaches will be instrumental in shaping future healthcare paradigms."

Reviewer #3:

Remarks to the Authors:

This manuscript investigates the value of environmental-clinical risk scores (ECRS) for predicting three types of health outcomes: mental health, cardiometabolic syndrome, and respiratory health. The study uses data from the HELIX project that includes data from six different European longitudinal birth cohorts. The results suggest that ECRS captures 12%, 51%, and 3% of the variance in mental, cardiometabolic, and respiratory health respectively. Predictive methods employed were LASSO, XGBoost, and Random Forests.

While the study is interesting, there are several major concerns about the manuscript in its current form. The modeling aspects are poorly written and organized. The manuscript fails to make effective use of figures and tables to clarify its methodology. Several of these issues as well as other issues are pointed out below.

We appreciate the reviewer taking time to analyze our paper and providing useful criticism to improve it. We clarified the Methods – modeling section and redid **Figure 2** (analysis workflow) to be easier to understand. We also added a new figure to show the availability of data in each cohort and performed a cohort-based sensitivity analysis. Further details are written below.

1. A detailed data table is needed to identify the multiple cohorts and indicate the variable types available for each cohort, which type of mental health outcome they are being used to predict, and what percent of the data is incomplete.

Following the reviewer suggestion, we added a supplementary table (see **Supplementary Table 3.5**) that outlines the data available for each cohort. The global list of variables used in the study, the % of missing values and other information are stated in **Supplementary Tables 3.2-3.4**. References to those tables are in the manuscript. Mental health clinical factors (and others) used as predictors are listed in **Supplementary Table 3.4**.

2. It take a lot of effort to understand that effectively there are three different prediction problems with mental health, cardiometabolic syndrome, and respiratory health outcomes as outputs and environmental factors as inputs. Fig. 2 which describes workflow fails to clearly point out what the model inputs and outputs are.

We thank the reviewer for this pertinent remark. We revised **Figure 2** to explicitly display the inputs and outcomes of the ECRS.

3. On the one hand, the paper treats ECRS like a simple scalar measure similar to PRS that can be used to assess environmental risk given a fixed health outcome. On the other hand, the environmental parameters seem to be inputs to the models. If multivariate environmental

inputs are used, where does the ECRS come in and how is it computed? All classification models presented here take in multiple inputs and try to predict a single health outcome as an output. Then where does a single ECRS variable corresponding to each outcome come in?

We thank the reviewer for pointing out that some of our main messages were not clear to our readership. We clarified the Methods - modeling section (lines 381-436) and provides a summary here: Our models compute several ECRS (simple scalar risk measures) by using several environmental and clinical predictors. These ECRS are computed through supervised machine learning models, which predict simple scalar measures as outcomes (MetS, P-factor, and lung function) using multiple environmental and clinical predictors (the full list is available in **Supplementary Table 3.2-3.4**). Training is first performed in a 10-fold cross validation procedure, where hyperparameters of the method are optimized and performances measured. Then final ECRS are computed using the optimized hyperparameters on the entire dataset.

4. Equally unclear are the roles of PRS, omics, and clinical variables. Are these model inputs? How to they relate to the ECRS?

Following the reviewer doubts, we better clarified these aspects in the paper (lines 382-385). Metabolites/proteins and clinical factors are utilized as predictors to compute ECRS and PRS are only used for risk stratification purposes.

5. LASSO is repeatedly referred to as a linear technique. LASSO could use a linear or generalized linear model. But due to the L1 constraint, LASSO estimation itself is not a linear approach. This needs to be clarified.

While it is true that the process of estimating the coefficients is not linear, we consider our LASSO models to be linear in the sense that they model the relationships between the input features and the output using a linear equation. We clarified it in the text (lines 396-399, in Methods, Statistical analysis, part 2. Modeling) as such:

"Note that technically, while the model itself is linear (it models the relationships between the input features and the output using a linear equation), the optimization process due to the L1 penalty is non-linear. One of its main advantages is its ability of handling high dimensional data through regularization."

6. Since the HELIX study includes data from six different cohorts, it is surprising that the cohort index was not treated as a covariate to compensate for cohort-specific bias. Not only to the cohort indices need to be treated as covariates the model, the authors should examine performance differences and interpretation differences that may exist across cohorts/data sources.

To specifically address cohort bias, we employed a two-step modeling approach, as described in the Method section (lines 423-436), to penalize contributions of features strongly correlated with cohorts (and hence, untrustworthy). Consequently, both predictive performances and extracted relationships were adjusted for the cohort bias.

Following the second part of the reviewer's comment, we performed a cohort-based sensitivity analysis for XGBoost models, where we iteratively trained those on five cohorts to predict the

sixth (using the same hyperparameters as before) (see Methods, lines 478-484; Results, lines 693-701; and Discussion, lines 706-708). We thank the reviewer for this relevant remark.

References:

1. Bergstra, J., Bardenet, R., Bengio, Y. & Kégl, B. Algorithms for Hyper-Parameter Optimization. in *Advances in Neural Information Processing Systems* vol. 24 (Curran Associates, Inc., 2011).
2. Vrijheid, M. *et al.* Advancing tools for human early lifecourse exposome research and translation (ATHLETE). *Environ. Epidemiol.* **5**, e166 (2021).
3. Hastie, T., Tibshirani, R. & Friedman, J. Overview of Supervised Learning. in *The Elements of Statistical Learning: Data Mining, Inference, and Prediction* (eds. Hastie, T., Tibshirani, R. & Friedman, J.) 9–41 (Springer, 2009). doi:10.1007/978-0-387-84858-7_2.
4. Hastie, T., Tibshirani, R. & Friedman, J. Model Assessment and Selection. in *The Elements of Statistical Learning: Data Mining, Inference, and Prediction* (eds. Hastie, T., Tibshirani, R. & Friedman, J.) 219–259 (Springer, 2009). doi:10.1007/978-0-387-84858-7_7.
5. Neufcourt, L. *et al.* Assessing How Social Exposures Are Integrated in Exposome Research: A Scoping Review. *Environ. Health Perspect.* **130**, 116001 (2022).
6. Balagopal, P. (Babu) *et al.* Nontraditional Risk Factors and Biomarkers for Cardiovascular Disease: Mechanistic, Research, and Clinical Considerations for Youth. *Circulation* **123**, 2749–2769 (2011).
7. Agier, L. *et al.* Early-life exposome and lung function in children in Europe: an analysis of data from the longitudinal, population-based HELIX cohort. *Lancet Planet. Health* **3**, e81–e92 (2019).
8. Murray, G. K. *et al.* Could Polygenic Risk Scores Be Useful in Psychiatry?: A Review. *JAMA Psychiatry* **78**, 210–219 (2021).
9. Caspi, A. *et al.* The p Factor: One General Psychopathology Factor in the Structure of Psychiatric Disorders? *Clin. Psychol. Sci. J. Assoc. Psychol. Sci.* **2**, 119–137 (2014).

10. Finsaas, M. C., Bufferd, S. J., Dougherty, L. R., Carlson, G. A. & Klein, D. N. Preschool psychiatric disorders: homotypic and heterotypic continuity through middle childhood and early adolescence. *Psychol. Med.* **48**, 2159–2168 (2018).
11. Rutter, M., Kim-Cohen, J. & Maughan, B. Continuities and discontinuities in psychopathology between childhood and adult life. *J. Child Psychol. Psychiatry* **47**, 276–295 (2006).
12. Wray, N. R. *et al.* From Basic Science to Clinical Application of Polygenic Risk Scores: A Primer. *JAMA Psychiatry* **78**, 101–109 (2021).
13. Shah, J. L., Jones, N., Os, J. van, McGorry, P. D. & Gülöksüz, S. Early intervention service systems for youth mental health: integrating pluripotentiality, clinical staging, and transdiagnostic lessons from early psychosis. *Lancet Psychiatry* **9**, 413–422 (2022).

REVIEWERS' COMMENTS:

Reviewer #1 (Remarks to the Author):

The authors responded to all the criticisms. Thanks

Reviewer #2 (Remarks to the Author):

I would like to commend the authors on a thorough and clear discussion of the critiques. All of my previous concerns are satisfied and I have no remaining comments.

Reviewer #3 (Remarks to the Author):

The authors have addressed all prior concerns satisfactorily.